# MoORE: SVD-based Model MoE-ization for Conflict- and Oblivion-Resistant Multi-Task Adaptation

**Shen Yuan**[1,2]  **Yin Zheng**[2]  **Taifeng Wang**[2]  **Binbin Liu**[2]  **Hongteng Xu**[1,3,4*]

[1]Gaoling School of Artificial Intelligence, Renmin University of China  [2]ByteDance
[3]Beijing Key Laboratory of Research on Large Models and Intelligent Governance
[4]Engineering Research Center of Next-Generation Intelligent Search and Recommendation, MOE

## Abstract

Adapting large-scale foundation models in multi-task scenarios often suffers from task conflict and oblivion. To mitigate such issues, we propose a novel "model MoE-ization" strategy that leads to a conflict- and oblivion-resistant multi-task adaptation method. Given a weight matrix of a pre-trained model, our method applies SVD to it and introduces a learnable router to adjust its singular values based on tasks and samples. Accordingly, the weight matrix becomes a Mixture of Orthogonal Rank-one Experts (MoORE), in which each expert corresponds to the outer product of a left singular vector and the corresponding right one. We can improve the model capacity by imposing a learnable orthogonal transform on the right singular vectors. Unlike low-rank adaptation (LoRA) and its MoE-driven variants, MoORE guarantees the experts' orthogonality and maintains the column space of the original weight matrix. These two properties make the adapted model resistant to the conflicts among the new tasks and the oblivion of its original tasks, respectively. Experiments on various datasets demonstrate that MoORE outperforms existing multi-task adaptation methods consistently, showing its superiority in terms of conflict- and oblivion-resistance. The code is available at `https://github.com/DaShenZi721/MoORE`.

## 1 Introduction

Parameter-efficient adaptation/fine-tuning [71] plays a central role in the practical deployment of large-scale pre-training models, especially in multi-task learning (MTL) scenarios [80, 61, 11, 62, 42]. Take large language models (LLMs) [82] in natural language processing (NLP) tasks as an example. To increase intrinsic knowledge and maintain generalization power, a pre-trained LLM often needs to learn multiple downstream tasks in different domains simultaneously or sequentially. To achieve multi-task adaptation, parameter-efficient fine-tuning (PEFT) methods like low-rank adaptation (LoRA) [25] and its variants [68, 2, 60, 65, 31] have been proposed. However, the practical applications of these methods are often limited because they often suffer from **task conflict and oblivion**: $i$) The model adapted for one task often leads to the performance degradation in other tasks (also known as negative transfer [70, 79] or destructive interference [11]). $ii$) Learning new tasks may result in catastrophic forgetting [62] — the model performance deteriorates severely in previously learned tasks.

Essentially, task conflict and oblivion arise because the diversity of different tasks requires the models to adapt their parameters in different directions [72, 74, 13, 26, 15]. Some recent methods [31, 30, 60] combine the Mixture-of-Experts (MoE) architecture [27, 55, 29] with PEFT, mitigating the interferences across different tasks by activating task-specific parameters. Given a layer of a pre-trained model, these methods learn multiple adapters associated with a router in multi-task scenarios.

---

*Corresponding author. Email: hongtengxu313@gmail.com

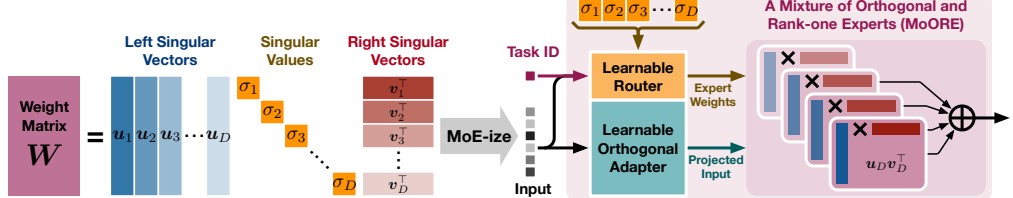

(a) An illustration of our SVD-based model MoE-ization strategy and MoORE architecture

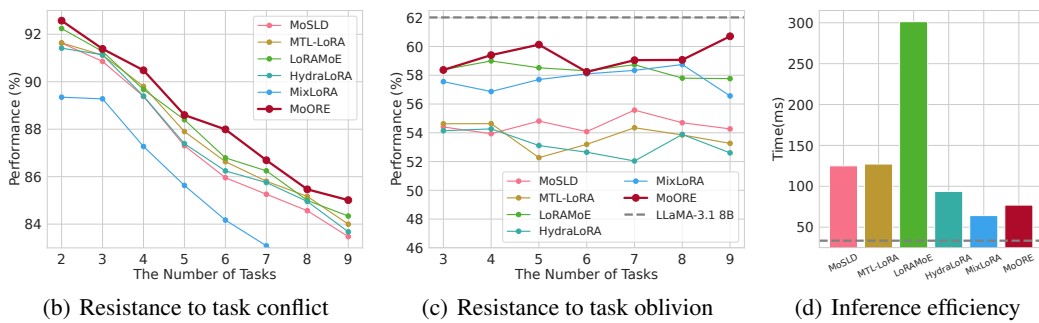

(b) Resistance to task conflict     (c) Resistance to task oblivion     (d) Inference efficiency

Figure 1: (a) An illustration of our model MoE-ization strategy and the corresponding MoORE architecture. (b) The comparison for various multi-task adaptation methods on fine-tuning LLaMA-3.1 8B [20] on the CSR-MTL constructed by nine tasks [9, 8, 45, 5, 54, 77, 53, 59]. MoORE consistently works better than the baselines when the number of tasks is larger than one. (c) Before adaptation, LLaMA-3.1 8B achieves encouraging overall performance (i.e., the gray dashed line) in seven tasks [24, 23, 83, 58, 51, 7, 4, 10]. MoORE mitigates the performance degradation and outperforms the most competitive baseline LoRAMoE [14]. (d) MoORE's runtime is comparable to that of its competitors. Compared to the original LLaMA-3.1 8B (i.e., the gray dashed line), MoORE increases the inference time moderately.

Each adapter may inherit task-specific knowledge, and the router selects/fuses these adapters based on the input data and tasks.

In principle, we call the above strategy "**Model MoE-ization**" because it converts the original linear layer to an MoE architecture. In this study, we propose a novel model MoE-ization strategy, with the help of the singular value decomposition (SVD), leading to a conflict- and oblivion-resistant multi-task adaptation method. As illustrated in Figure 1(a), given a weight matrix of a pre-trained model, our method applies SVD [1] to it and introduces a learnable router to adjust its singular values based on tasks and samples. Accordingly, the weight matrix becomes a Mixture of Orthogonal Rank-one Experts (**MoORE**), where each expert is constructed by the outer product of a left singular vector and the corresponding right one. To further improve the capacity of MoORE, we impose a learnable orthogonal adapter, which is implemented by a Householder reflection adaptation module [75].

Unlike existing methods, which learn some additional low-rank experts without any constraints on their relations, MoORE extracts many rank-one experts intrinsically from the SVD of the original weight matrix. Such a design *imposes the orthogonality of the experts*, avoiding their information redundancy and undesired interferences. Moreover, similar to existing orthogonal fine-tuning strategies [49, 75], MoORE *maintains the column space of the original weight matrix* and thus inherit the pre-training ability better. Thanks to the above two properties, MoORE consistently outperforms SOTA methods in various multi-task adaptation scenarios. The results in Figure 1 show the superiority of MoORE in mitigating task conflict and oblivion and its competitive inference efficiency.

## 2 Related Work and Preliminaries

### 2.1 Adapter-based Methods for Multi-Task Adaptation

Multi-task adaptation aims to fine-tune a pre-trained foundation model simultaneously or sequentially in multiple downstream tasks [34, 32]. Focusing on this problem, many adapter-based methods have been proposed [37, 50, 57, 65, 40]. In particular, denote the data of $K$ tasks as $\{\mathcal{D}_k\}_{k=1}^K$. Given a

pre-trained foundation model, whose parameters are denoted as $\boldsymbol{\theta}$, multi-task adaptation can often be formulated in the framework of maximum likelihood estimation:

$$\max_{\Delta\boldsymbol{\theta}} \sum\nolimits_{k=1}^{K} P_{\boldsymbol{\theta}\cup\Delta\boldsymbol{\theta}}(\mathcal{D}_k), \tag{1}$$

where $P_{\boldsymbol{\theta}\cup\Delta\boldsymbol{\theta}}(\mathcal{D}_k)$ denotes the likelihood of the $k$-th task's data, which is parameterized by the original model parameters $\boldsymbol{\theta}$ and the adapters' parameters $\Delta\boldsymbol{\theta}$.

Unlike learning and ensembling different models, adapter-based methods improve the overall model performance by sharing parameters and domain knowledge across various tasks, leading to moderate increases in parameters and complexity. For instance, Hyperformer [44] utilizes a shared hyper-network to generate task-specific adapters, reducing the number of learnable parameters. Recently, some methods extend LoRA [25] to multi-task adaptation, e.g., MultiLoRA [68], MTL-LoRA [73], HydraLoRA [60], and so on, which learns multiple low-rank adapters to handle diverse tasks.

## 2.2 Connections to MoE Architectures

The Mixture-of-Experts (MoE) architecture was initially introduced by the work in [27], which is constructed by multiple specialized networks (called experts) and a router. Given an input data $\boldsymbol{x} \in \mathcal{X}$, the MoE derives the output $\boldsymbol{y} \in \mathcal{Y}$ as $\boldsymbol{y} = \sum_{m=1}^{M} g_m(\boldsymbol{x})f_m(\boldsymbol{x})$, where $f_m : \mathcal{X} \mapsto \mathcal{Y}$ denotes the $m$-th expert, which achieves a mapping from the sample space $\mathcal{X}$ to the output space $\mathcal{Y}$. $g : \mathcal{X} \mapsto \mathbb{R}^M$ denotes the router, and $g_m$ denotes its $m$-th output. The router adjusts the experts' significance based on input data. When applying a sparse routing strategy, i.e., activating only a few experts for each input [55, 29, 17, 3, 16, 46], the MoE architecture supports building large-scale models while maintaining computational efficiency. Due to its advantages, many large language models, e.g., DeepSeek [12], Grok3, and Qwen3[2], are built based on MoE, and many efforts have been made to convert well-trained dense models into MoE architectures [28, 22, 52, 47, 78].

As mentioned before, most existing adapter-based multi-task adaptation methods actually "MoE-ize" pre-trained models. Given a weight matrix $\boldsymbol{W}$ and its input $\boldsymbol{x}$, these methods apply multiple low-rank adapters as experts [76, 14, 35, 41] and add them to the pre-trained models, i.e.,

$$\boldsymbol{y} = \boldsymbol{W}\boldsymbol{x} + \sum\nolimits_{m=1}^{M} g_m(\boldsymbol{x})\boldsymbol{B}_m\boldsymbol{A}_m\boldsymbol{x}, \tag{2}$$

where $\boldsymbol{A}_m$ and $\boldsymbol{B}_m$ are low-rank matrices constructing the $m$-th expert $f_m$. For the router $g(\boldsymbol{x})$, some attempts have been made to develop advanced routing strategies, e.g., the dynamic routing in AdaMoLE [38] and the token-task hybrid routing in HMoRA [31]. Recently, some new MoE architectures have been proposed, including the asymmetric "Hydra" structure in HydraLoRA [60], the LEMoE [66] for lifelong model editing, and the OMoE [18] for orthogonal output. However, the above methods rely purely on data-driven strategies to determine the experts' functionality and domain knowledge. Without necessary regularization on the relations across different experts, the experts often suffer from the load imbalance issue, i.e., a limited number of experts are over-trained and applied for multiple tasks, while many experts are seldom used. This issue harms the capacity and generalizability of the models, increasing the risk of task conflict and oblivion.

# 3 Proposed Method

In this study, we propose a new model MoE-ization method for multi-task adaptation. In principle, our method imposes orthogonality on the experts and further regularizes their output spaces, which helps mitigate task conflict and oblivion.

## 3.1 SVD-based Model MoE-ization

Consider a pre-trained weight matrix $\boldsymbol{W} \in \mathbb{R}^{D_{\text{out}} \times D}$. Without loss of generality, we assume $D_{\text{out}} \geq D$ and $\text{Rank}(\boldsymbol{W}) = D$. The SVD of the matrix is denoted as

$$\boldsymbol{W} = \boldsymbol{U}\text{diag}(\boldsymbol{\sigma})\boldsymbol{V}^\top = \sum\nolimits_{d=1}^{D} \underbrace{\sigma_d}_{\text{weight}} \cdot \underbrace{(\boldsymbol{u}_d\boldsymbol{v}_d^\top)}_{\text{expert}}, \tag{3}$$

---

[2]https://github.com/QwenLM/Qwen3

where $\boldsymbol{U} = [\boldsymbol{u}_1, \cdots, \boldsymbol{u}_D] \in \mathbb{R}^{D_{\text{out}} \times D}$ contains left singular vectors, $\boldsymbol{V} = [\boldsymbol{v}_1, \cdots, \boldsymbol{v}_D] \in \mathbb{R}^{D \times D}$ contains right singular vectors, and $\boldsymbol{\sigma} = [\sigma_1, \cdots, \sigma_D]^\top$ is a vector of singular values.

As shown in (3), there exists an intrinsic but static MoE architecture hidden in the SVD of the weight matrix — each $\boldsymbol{W}$ corresponds to the mixture of $D$ orthogonal and rank-one experts, in which the $d$-th expert is the outer product of $\boldsymbol{u}_d$ and $\boldsymbol{v}_d$ and its weight is fixed as $\sigma_d$. Motivated by this intrinsic MoE, our method derives the proposed MoORE model for multi-task adaptation, which reuses the experts while introducing the following two modifications:

- **A hybrid routing strategy:** Inspired by HMoRA [31], we adjust the experts' weights according to input data and tasks, leading to a hybrid routing strategy. Given an input of the $k$-th task, denoted as $\boldsymbol{x}^{(k)} \in \mathbb{R}^d$, we determines the weight of the $d$-th expert as

$$g_d(\boldsymbol{x}^{(k)}) = \underbrace{\boldsymbol{p}_d^\top \boldsymbol{t}_k}_{\text{task-level}} + \underbrace{\boldsymbol{q}_d^\top \boldsymbol{\Gamma} \boldsymbol{x}^{(k)}}_{\text{sample-level}}, \tag{4}$$

where $\boldsymbol{t}_k \in \mathbb{R}^{D_t}$ denotes the embedding of the $k$-th task, and the vector $\boldsymbol{p}_d \in \mathbb{R}^{D_t}$ projects the task embedding to a task-level weight. Similarly, applying the matrix $\boldsymbol{\Gamma} \in \mathbb{R}^{D_s \times D}$ and the vector $\boldsymbol{q}_d \in \mathbb{R}^{D_s}$, we project the input $\boldsymbol{x}^{(k)}$ to a sample-level weight. In practice, we set $D_s, D_t \ll D$ to reduce the router's parameters and computational cost. The final weight $g_d(\boldsymbol{x}^{(k)})$ is the summation of the task- and sample-level weights.

- **An orthogonal adapter of input:** To further increase the model capacity, we can apply a learnable orthogonal transform to the input, i.e., $\boldsymbol{H}\boldsymbol{x}$, where the learnable orthogonal transform $\boldsymbol{H}$ can be implemented efficiently by the butterfly orthogonal fine-tuning (BOFT) module in [36], the Givens rotation adapter [43], or the Householder reflection adapter (HRA) [75]. In this study, we implement $\boldsymbol{R}$ by HRA, which corresponds to the product of $L$ Householder reflections, i.e.,

$$\boldsymbol{H} = \prod_{\ell=1}^{L} \left( \boldsymbol{I} - \frac{1}{\|\boldsymbol{r}_\ell\|_2^2} \boldsymbol{r}_\ell \boldsymbol{r}_\ell^\top \right), \tag{5}$$

whose learnable parameters are $\boldsymbol{R} = [\boldsymbol{r}_1, \cdots, \boldsymbol{r}_L] \in \mathbb{R}^{D \times L}$. Note that applying orthogonal adapters maintains the angles between neurons (i.e., the rows of the weight matrix $\boldsymbol{W}$), which helps preserve the knowledge of the pre-trained model when enhancing model capacity [49, 36, 75].

Applying the above SVD-based model MoE-ization strategy, we obtain the proposed MoORE model, which introduces $D$ orthogonal and rank-one experts into the model and encodes the input data as

$$\boldsymbol{y} = \boldsymbol{W}\boldsymbol{H}\boldsymbol{x}^{(k)} + \sum_{d=1}^{D} \underbrace{g_d(\boldsymbol{x}^{(k)})}_{\text{router}} \underbrace{(\boldsymbol{u}_d \boldsymbol{v}_d^\top \boldsymbol{H})}_{\text{expert}} \boldsymbol{x}^{(k)} = \boldsymbol{U}\text{diag}(g(\boldsymbol{x}^{(k)}) + \boldsymbol{\sigma})\boldsymbol{V}^\top \boldsymbol{H}\boldsymbol{x}^{(k)}, \tag{6}$$

where $g(\boldsymbol{x}^{(k)}) = \boldsymbol{P}^\top \boldsymbol{t}_k + \boldsymbol{Q}^\top \boldsymbol{\Gamma} \boldsymbol{x}^{(k)} \in \mathbb{R}^D$.

The learnable parameters of MoORE include $\boldsymbol{T} = [\boldsymbol{t}_k] \in \mathbb{R}^{D_t \times K}$, $\boldsymbol{P} = [\boldsymbol{p}_d] \in \mathbb{R}^{D_t \times D}$, $\boldsymbol{Q} = [\boldsymbol{q}_d] \in \mathbb{R}^{D_s \times D}$, $\boldsymbol{\Gamma} \in \mathbb{R}^{D_s \times D}$, and $\boldsymbol{R} \in \mathbb{R}^{D \times L}$. As shown in (6), given the SVD of $\boldsymbol{W}$ (which can be computed in advance), we can implement MoORE with low complexity via the following steps:

$$1)\ \boldsymbol{z} = \underbrace{\boldsymbol{V}\boldsymbol{H}\boldsymbol{x}^{(k)}}_{\mathcal{O}(D(L+D))}, \quad 2)\ g(\boldsymbol{x}^{(k)}) = \underbrace{\boldsymbol{P}^\top \boldsymbol{t}_k}_{\mathcal{O}(DD_t)} + \underbrace{\boldsymbol{Q}^\top \boldsymbol{\Gamma} \boldsymbol{x}^{(k)}}_{\mathcal{O}(DD_s)}, \quad 3)\ \boldsymbol{y} = \underbrace{\boldsymbol{U}(g(\boldsymbol{x}^{(k)}) + \boldsymbol{\sigma})\boldsymbol{z}}_{\mathcal{O}(DD_{\text{out}})}. \tag{7}$$

As shown in (7), the overall complexity of MoORE is $\mathcal{O}(D(D_{\text{out}} + D + D_t + D_s + L))$. In practice, we set $L, D_t, D_s \ll \min\{D, D_{\text{out}}\}$ to reduce the complexity. Moreover, we can merge the orthogonal adapter into each expert in the inference phase, i.e., obtaining $\boldsymbol{V}' = \boldsymbol{V}^\top \boldsymbol{H}$, and the complexity further reduces to $\mathcal{O}(D(D_{\text{out}} + D + D_t + D_s))$. Figure 1(d) shows that the inference efficiency of MoORE is comparable to its competitors.

## 3.2 Comparisons with Existing MoE-based Multi-Task Adaptation Methods

Our SVD-based model MoE-ization method provides a new technical route for multi-task adaptation: **Instead of learning an MoE with few strong extrinsic experts, our method constructs an MoE with many simple but structured experts intrinsically based on the pre-trained weight matrix.** Tables 1 and 2 compare different methods on their MoE architectures, theoretical properties, and computational efficiency, respectively.

Table 1: Comparisons for various MoE-based multi-task adaptation methods on their implementations and properties, where $M$ ($D$ in our method) is the number of experts, $k$ is the task index, GS$(\cdot)$ denotes the Gram-Schmidt orthogonalization [6].

| Method | Router | Experts | #Experts | Rank of Experts | Orthogonality of Experts | Maintaining Range($\boldsymbol{W}$) |
|---|---|---|---|---|---|---|
| LoRAMoE [14] | Softmax($\boldsymbol{Sx}$) | $\{\boldsymbol{B}_m\boldsymbol{A}_m\}_{m=1}^M$ | $M$ | $r$ | No | No |
| MixLoRA [30] | Top$_2$(Softmax($\boldsymbol{Sx}$)) | $\{\boldsymbol{B}_m\boldsymbol{A}_m\}_{m=1}^M$ | $M$ | $r$ | No | No |
| MoSLD [81] | Softmax($\boldsymbol{Sx}$) | $\{\boldsymbol{B}\boldsymbol{A}_m\}_{m=1}^M$ | $M$ | $r$ | No | No |
| HydraLoRA [60] | Softmax($\boldsymbol{Sx}$) | $\{\boldsymbol{B}_m\boldsymbol{A}\}_{m=1}^M$ | $M$ | $r$ | No | No |
| MTL-LoRA [73] | Softmax($\boldsymbol{\phi}_k$) | $\{\boldsymbol{B}_m\boldsymbol{\Lambda}_k\boldsymbol{A}\}_{m=1,k=1}^{M,K}$ | $MK$ | $r$ | No | No |
| OMoE [18] | Softmax($\boldsymbol{Sx}$) | GS($\{\boldsymbol{B}_m\boldsymbol{A}_m\}_{m=1}^M$) | $M$ | $r$ | Yes | No |
| **MoORE (Ours)** | $\boldsymbol{P}^\top\boldsymbol{t}_k + \boldsymbol{Q}^\top\boldsymbol{\Gamma x}$ | $\{\boldsymbol{u}_d\boldsymbol{v}_d^\top\boldsymbol{H}\}_{d=1}^D$ | $D$ | $1$ | Yes | Yes |

- **The design of router.** Given an input $\boldsymbol{x} \in \mathbb{R}^D$, most existing methods leverage a sample-level routing strategy. Typically, they apply a linear projection $\boldsymbol{S} \in \mathbb{R}^{M \times D}$ to it and pass the projection result through a softmax operator, leading to a nonnegative and normalized weight vector for $M$ experts. Among them, MixLoRA [30] further applies a sparse routing mechanism — for each input, it only activates the two experts that correspond to the top-2 weights. Instead, MTL-LoRA [73] leverages a task-level routing strategy, which determines the experts' weights by passing a task-specific embedding (i.e., $\boldsymbol{\phi}_k \in \mathbb{R}^M$, $k \in \{1, \cdots, K\}$ indicates the task index) through a softmax operator. Unlike existing methods, our MoORE considers the task- and sample-level information jointly. The advantage of this hybrid routing strategy is that when the same sample serves for different tasks, MoORE can assign task-specific weights to the experts and thus leverage different domain knowledge accordingly.

- **The design of experts.** Most existing methods apply $M$ low-rank adapters as experts, i.e., $\{\boldsymbol{B}_m\boldsymbol{A}_m\}_{m=1}^M$, where $\boldsymbol{B} \in \mathbb{R}^{D_{\text{out}} \times r}$ and $\boldsymbol{A} \in \mathbb{R}^{r \times D}$ are two rank-$r$ matrices. To reduce the number of learnable parameters, some methods, e.g., MoSLD [81], HydraLoRA [60], and MTL-LoRA [73], reuse the same $\boldsymbol{A}$ or $\boldsymbol{B}$ for all experts. MTL-LoRA [73] further introduces a task-specific matrix $\boldsymbol{\Lambda}_k$ for each expert, where $k = 1, ..., K$ indicates the task index. As a result, it creates $MK$ low-rank experts and activates $M$ experts per task. Our MoORE contains $D$ orthogonal rank-one experts $\{\boldsymbol{u}_d\boldsymbol{v}_d^\top\boldsymbol{H}\}_{d=1}^D$, in which the orthogonal adapter $\boldsymbol{H}$ is shared by all experts. Unlike existing methods, MoORE applies many simple but structured experts. Such a design has several advantages:

  1. **Imposing orthogonality for mitigating task conflict:** The experts of MoORE are orthogonal to each other because $\boldsymbol{u}_d^\top\boldsymbol{u}_{d'} = 0$ for all $d \neq d'$. The orthogonality ensures that the experts have different functionalities and domain knowledge, without redundant information. In particular, by activating different experts for different tasks, MoORE suppresses the interferences across the tasks in the training phase, thus mitigating the task conflict issue. It is worth noting that O-LoRA [67] also introduces orthogonality constraints within its LoRA adapters. However, there are two notable differences between O-LoRA and MoORE: $i$) O-LoRA uses a regularization term to pursue orthogonality, which can not strictly make its LoRA adapters orthogonal to each other. $ii$) O-LoRA does not adopt a MoE architecture. To deal with different tasks, O-LoRA adds new LoRA adapters for each sequentially incoming task, rather than designing and learning a flexible routing mechanism for given and fixed experts. As a result, the model complexity of O-LoRA is linear to the number of tasks. Among existing MoE-based methods, OMoE [18] is the only one imposing orthogonality on experts. However, it applies Gram-Schmidt orthogonalization algorithm [6] to the concatenation of $M$ experts' output, i.e., GS($[\boldsymbol{B}_1\boldsymbol{A}_m\boldsymbol{x}, \cdots, \boldsymbol{B}_M\boldsymbol{A}_m\boldsymbol{x}]$). As a result, imposing orthogonality requires additional $\mathcal{O}(D_{\text{out}}M^2)$ operations per sample, which is less efficient than ours.

  2. **Maintaining Range($\boldsymbol{W}$) for mitigating task oblivion:** The column space of each expert, i.e., Range($\boldsymbol{u}_d\boldsymbol{v}_d^\top\boldsymbol{H}$), is the same with Range($\boldsymbol{u}_d$). Accordingly, the output space of MoORE is the direct sum of $\{\text{Range}(\boldsymbol{u}_d)\}_{d=1}^D$, which is the same as $\boldsymbol{W}$'s column space, i.e.,

$$\bigoplus_{d=1}^D \text{Range}(\boldsymbol{u}_d\boldsymbol{v}_d^\top\boldsymbol{H}) = \bigoplus_{d=1}^D \text{Range}(\boldsymbol{u}_d) = \text{Range}(\boldsymbol{U}) = \text{Range}(\boldsymbol{W}). \tag{8}$$

Table 2: Comparisons for various MoE-based multi-task adaptation methods on their computational efficiency.

| Method | #Learnable Parameters | | Computational Complexity |
| --- | --- | --- | --- |
| | Router | Experts | |
| LoRAMoE [14] | $MD$ | $M(D_{\text{out}} + D)r$ | $\mathcal{O}(D_{\text{out}}D + M(D + D_{\text{out}}r + Dr))$ |
| MixLoRA [30] | $MD$ | $M(D_{\text{out}} + D)r$ | $\mathcal{O}(D_{\text{out}}D + MD + (D_{\text{out}}r + Dr))$ |
| MoSLD [81] | $MD$ | $(D_{\text{out}} + MD)r$ | $\mathcal{O}(D_{\text{out}}D + D_{\text{out}}r + M(Dr + D))$ |
| HydraLoRA [60] | $MD$ | $(MD_{\text{out}} + D)r$ | $\mathcal{O}(D_{\text{out}}D + Dr + M(D_{\text{out}}r + D))$ |
| MTL-LoRA [73] | $MK$ | $(MD_{\text{out}} + Kr + D)r$ | $\mathcal{O}(D_{\text{out}}D + Dr + MD_{\text{out}}r + r^2)$ |
| OMoE [18] | $MD$ | $M(D_{\text{out}} + D)r$ | $\mathcal{O}(D_{\text{out}}(D + M^2) + M(D + D_{\text{out}}r + Dr))$ |
| **MoORE (Ours)** | $D_t K + (D_t + 2D_s)D$ | $LD$ | $\mathcal{O}((D_{\text{out}} + D + D_s + D_t + L)D)$ |

Table 3: Comparisons for various SVD-based fine-tuning methods on their implementations.

| Method | Model in Training | Learnable Parameters | Model in Inference |
| --- | --- | --- | --- |
| SVF [56] | $\boldsymbol{U}\text{diag}(\sigma)\boldsymbol{V}^\top$ | Nonparametric $\sigma$ | Original architecture: $\boldsymbol{W}_{\text{new}}$ |
| SVDiff [21] | $\boldsymbol{U}\text{diag}(\text{ReLU}(\sigma + \delta))\boldsymbol{V}^\top$ | Nonparametric $\delta$ | Original architecture: $\boldsymbol{W}_{\text{new}}$ |
| SVFT [33] | $\boldsymbol{U}(\text{diag}(\sigma) + \boldsymbol{M})\boldsymbol{V}^\top$ | Nonparametric $\boldsymbol{M}$ | Original architecture: $\boldsymbol{W}_{\text{new}}$ |
| KaSA [64] | $\boldsymbol{W}_{lr} + \Delta\boldsymbol{U}\text{diag}(\Delta\sigma)\Delta V^\top$ | Nonparametric $\Delta\boldsymbol{U}, \Delta\sigma, \Delta V^\top$ | Original architecture: $\boldsymbol{W}_{\text{new}}$ |
| SVFCL [69] | $\boldsymbol{U}\text{diag}(\sigma + \sum_{i=0}^t \sigma_i)\boldsymbol{V}^\top$ | Nonparametric $\sigma_i$'s | Original architecture: $\boldsymbol{W}_{\text{new}}$ |
| **MoORE (Ours)** | $\boldsymbol{W}\boldsymbol{H}$ $+\sum_{d=1}^D g_d(\boldsymbol{x}^{(k)})(\boldsymbol{u}_d\boldsymbol{v}_d^\top \boldsymbol{H})$ | Parametric $g(\cdot)$ and Householder reflections $\boldsymbol{H}$ | An MoE, where $\boldsymbol{W}\boldsymbol{H}$ and $\boldsymbol{v}_d^\top \boldsymbol{H}$ are pre-computed |

In single task adaptation scenarios, the work in [75] has shown that maintaining the column space of the weight matrix makes the adapted model inherit the ability of the pre-trained model better, mitigating the oblivion of the previously pre-trained task. In our work, we find that such maintenance is helpful in multi-task adaptation scenarios as well.

- **Computational efficiency.** Table 2 shows each method's learnable parameters and computational complexity. For the methods using the sample-based routing strategy, their routers contain $MD$ learnable parameters. Given a sample, the complexity of the router is $\mathcal{O}(MD)$. For the method [73] using the task-based routing strategy, its router is lightweight, containing $MK$ learnable parameters and determining the weights of its experts with complexity $\mathcal{O}(M)$. For the experts, using $M$ rank-$r$ experts leads to the complexity $\mathcal{O}(M(D_{\text{out}} + D)r)$. Reusing $\boldsymbol{A}$ or $\boldsymbol{B}$ (e.g., MoSLD [81] and HydraLoRA [60]) and applying sparse routing (e.g., MixLoRA [30]) can reduce the computational complexity significantly. In contrast, introducing additional parameters (e.g., the $\boldsymbol{\Lambda}_k$ in MTL-LoRA [73]) or operations (e.g., the Gram-Schmidt orthogonalization in OMoE [18]) leads to higher complexity. Existing methods construct an MoE with $M$ rank-$r$ experts, while our MoORE is an MoE with $D$ rank-one experts, whose number of experts is determined by the input dimension and thus much larger than $M$. To improve the computational efficiency of MoORE, we set $D_s$ and $D_t$ comparable to the rank $r$ and set $L$ comparable to $M$, respectively. As a result, the number of learnable parameters and the complexity of MoORE become comparable to most existing methods.

### 3.3 Comparisons with Existing SVD-based Fine-Tuning Methods

Our method employs SVD to perform model MoE-ization, constructing a complete MoE architecture. This fundamentally differs from existing SVD-based fine-tuning methods [48, 56, 21, 33, 64, 69]. To more clearly illustrate the differences between MoORE and these methods, we select five recent ones for comparison, including SVF [56], SVDiff [21], SVFT [33], KaSA [64], and SVFCL [69]. Tables 3 and 4 compare different methods in terms of implementation and application, respectively.

- **In the aspect of architecture**, MoORE is the only method that learns a parametric router to adjust singular values, leading to an MoE architecture for inference. In MoORE, each expert is constructed by the outer product of singular vectors and the learnable Householder reflections.

- **In the aspect of application**, only MoORE and SVFCL consider multi-task adaptation. MoORE is applicable for both simultaneous and sequential multi-task adaptation, while SVFCL only

Table 4: Comparisons for various SVD-based fine-tuning methods on their applications.

| Method | Multi-Task Learning | Consider Task Conflict | Consider Task Oblivion |
|--------|---------------------|------------------------|------------------------|
| SVF [56] | No | No | No |
| SVDiff [21] | No | No | No |
| SVFT [33] | No | No | No |
| KaSA [64] | No | No | No |
| SVFCL [69] | Yes (incremental learning) | No | Partially discussed |
| **MoORE (Ours)** | Yes | Yes, by orthogonal experts | Yes, by maintaining Range($\boldsymbol{W}$) |

considers incremental learning. Moreover, MoORE makes the first attempt to connect task conflict and oblivion to the orthogonality of experts and the column space of the parameter matrix.

# 4 Experiments

To demonstrate the effectiveness of MoORE in multi-task adaptation, we apply three MTL datasets and conduct comprehensive experiments on them. Representative results are shown in Figure 1 and the following content. More experimental details, e.g., the basic information of datasets, hyperparameter settings, ablation studies, routing weight analysis, and numerical results associated with figures, are shown in the Appendix.

## 4.1 Implementation Details

**Base model and baselines.** In the following experiments, we utilize LLaMA-3.1 8B[3] [20] as the base model and adapt it by various multi-task adaptation methods. In particular, we compare MoORE with LoRA [25] and the methods incorporating low-rank adapters as MoEs, including LoRAMoE [14], MoSLD [81], MTL-LoRA [73], HydraLoRA [60], and MixLoRA [30]. We implement the MoE architectures of the baselines mainly based on their default settings. For a fair comparison, we modify some baselines' hyperparameters to make the number of learnable parameters comparable for each method. For MoORE, we MoE-ize all linear layers of LLaMA-3.1 8B, including the "$QKVO$" modules of attention layers and the weight matrices of FFNs.

**Two datasets for evaluating conflict-resistance.** We consider two MTL datasets for commonsense reasoning (CSR) and natural language understanding (NLU), respectively. The **CSR-MTL** dataset is constructed by nine tasks, including ARC-Challenge (ARC-C), ARC-Easy (ARC-E) [9], Open-BookQA (OBQA) [45], PIQA [5], SocialIQA (SIQA) [54], BoolQ [8], Hellaswag (HellaS) [77], Winogrande (WinoG) [53], and CommonsenseQA (CSQA) [59]. These tasks are widely used to evaluate LLMs on various commonsense reasoning challenges, ranging from genuine grade-school level science questions to physical commonsense reasoning. The **NLU-MTL** dataset consists of seven tasks from GLUE [63], including CoLA, SST-2, MRPC, QQP, MNLI, QNLI, and RTE. These tasks are applied to evaluate the natural language understanding capabilities of LLMs, including natural language inference, textual entailment, sentiment analysis, semantic similarity, and so on.

**One dataset for evaluating oblivion-resistance.** In addition, we construct one more dataset, called **OR-MTL**, for evaluating the oblivion-resistance of different methods. The dataset includes seven tasks, including MMLU [24, 23], IFEval [83], BIG-Bench Hard (BBH) [58], GPQA [51], HumanEval [7], MBPP [4], and GSM-8K [10]. The base model, LLaMA-3.1 8B, achieves encouraging performance in these tasks. After adapting it on CSR-MTL, we record the performance of the adapted models in OR-MTL and assess the ability of different methods to mitigate task oblivion.

**Experimental settings.** When adapting the pre-trained model on CSR-MTL and NLU-MTL, we set the training epoch to 2 and 5, respectively. The learning rate is set to $3 \times 10^{-4}$, with AdamW [39] as the optimizer. For CSR-MTL, we set the batch size to 8, whereas for NLU-MTL, we set the batch size to 64. Both training and testing are conducted on one NVIDIA A100 GPU.

---

[3] https://huggingface.co/meta-llama/Llama-3.1-8B-Instruct

Table 5: Results (%) of various methods on CSR-MTL. The best results on each dataset are shown in **bold**, and the second best results are shown in underline.

| Method | #Params | ARC-C | ARC-E | BoolQ | OBQA | PIQA | SIQA | HellaS | WinoG | CSQA | Overall |
|---|---|---|---|---|---|---|---|---|---|---|---|
| Full Fine-tuning | 100.00% | 80.12 | 87.92 | 74.56 | 87.60 | 88.96 | 80.55 | **95.91** | 81.69 | 81.16 | 84.27 |
| LoRA [25] | 2.09% | 77.56 | 85.77 | 70.43 | 81.60 | 82.97 | 76.00 | 93.00 | 71.11 | 77.40 | 79.54 |
| MixLoRA [30] | 3.00% | 79.18 | 87.50 | 72.02 | 86.60 | 87.38 | 78.86 | 93.65 | 77.35 | 80.43 | 82.55 |
| MoSLD [81] | 1.49% | 80.29 | 86.87 | 74.16 | 86.40 | 88.58 | 79.73 | 95.03 | 80.58 | 80.34 | 83.55 |
| HydraLoRA [60] | 2.72% | 79.27 | 88.09 | 74.34 | 85.60 | 88.96 | 79.84 | 95.36 | **83.03** | 80.10 | 83.84 |
| MTL-LoRA [73] | 2.69% | 80.55 | 88.38 | **75.26** | 85.80 | 88.41 | 80.45 | 95.57 | 81.61 | 80.75 | 84.09 |
| LoRAMoE [14] | 2.19% | 80.97 | 88.51 | 74.37 | 86.20 | 89.45 | 80.91 | 95.26 | 81.29 | 82.06 | 84.34 |
| MoORE $_{L=0}$ | 2.72% | **82.51** | **89.35** | 74.59 | 87.80 | 89.39 | 79.89 | 95.52 | 81.53 | 84.22 | 84.98 |
| MoORE $_{L=2}$ | 2.75% | 81.91 | **89.35** | 74.56 | 87.60 | **89.83** | 80.45 | 95.46 | 81.53 | 84.52 | 85.02 |
| MoORE $_{L=4}$ | 2.78% | 82.25 | 89.23 | 74.74 | **88.60** | **89.83** | 80.04 | 95.41 | 80.98 | 84.11 | 85.02 |
| MoORE $_{L=8}$ | 2.84% | 82.25 | 89.31 | 74.74 | 87.80 | 89.55 | 79.89 | 95.48 | 82.40 | **84.60** | 85.11 |

Table 6: Results (%) of various methods on NLU-MTL. The best results on each dataset are shown in **bold**, and the second best results are shown in underline. We report the matched accuracy for MNLI, Matthew's correlation for CoLA, and average correlation for STS-B.

| Method | #Params | CoLA | MNLI | MRPC | QNLI | QQP | RTE | SST-2 | Overall |
|---|---|---|---|---|---|---|---|---|---|
| LoRA [25] | 2.09% | 39.69 | 86.20 | 80.88 | 88.16 | 87.80 | 80.14 | 90.60 | 79.07 |
| MixLoRA [30] | 3.00% | 61.71 | 89.98 | 82.84 | 94.47 | 90.11 | 85.56 | 95.53 | 85.74 |
| MoSLD [81] | 1.49% | 58.55 | 90.34 | 85.29 | 94.95 | 90.08 | 89.53 | 96.33 | 86.44 |
| MTL-LoRA [73] | 2.69% | 60.65 | 90.26 | 87.01 | 95.26 | **91.95** | 90.98 | 96.10 | 87.46 |
| HydraLoRA [60] | 2.72% | 64.92 | 89.97 | 86.76 | 95.31 | 91.62 | 87.73 | 96.44 | 87.54 |
| LoRAMoE [14] | 2.19% | 62.84 | **91.01** | 88.97 | 95.55 | 91.01 | 90.61 | 96.56 | 88.08 |
| MoORE $_{L=0}$ | 2.72% | 69.09 | 89.86 | **91.91** | 94.44 | 90.73 | **91.34** | 96.22 | **89.08** |
| MoORE $_{L=2}$ | 2.75% | **69.23** | 90.35 | 89.70 | 95.37 | 90.75 | **91.34** | 96.56 | 89.04 |

## 4.2 Performance in Conflict-Resistance

Tables 5 and 6 compare MoORE with its competitors on CSR-MTL and NLU-MTL, respectively. Using a comparable number of learnable parameters for each dataset, MoORE achieves the best or comparable performance across all tasks and thus obtains the best average performance. These results demonstrate the superiority of MoORE in mitigating task conflict.

**The impact of orthogonal adapter.** In the experiment on CSR-MTL, we increase the number of Householder reflections in $H$ (i.e., $L$) from 0 to 8 and find that MoORE exhibits consistent improvements in performance. In the experiment on NLU-MTL, however, applying the orthogonal adapter may not improve performance. In our opinion, this phenomenon indicates that the commonsense reasoning tasks in CSR-MTL require more domain knowledge not covered by the pre-trained model. As a result, introducing the orthogonal adapter increases the number of learnable parameters and enhances the model capacity accordingly. In contrast, the text classification tasks in NLU-MTL rely more on the non-specific natural language knowledge captured by the pre-trained model. Therefore, without introducing more learnable parameters, adjusting the singular values of the pre-trained weight matrix is sufficient to achieve encouraging performance.

**Conflict-resistance regarding task number and difficulty.** To further compare and analyze the conflict-resistance capabilities of different methods, we conduct comparative experiments on CSR-MTL by varying the number and difficulty of the tasks. In particular, for each task in CSR-MTL, we first calculate the average of all the methods' performance based on the results in Table 5. The average performance of the methods in a task measures the difficulty of the task for the base model — the lower the average performance is, the more difficult the task is. Then, we sort the tasks in ascending order based on their difficulty. Finally, we adapt the base model for the top-$K$ tasks, $K = 2, ..., 9$, and show the performance of different methods in Figure 1(b). With the increase of task number and difficulty, all the methods suffer performance degradation because $i$) task conflict becomes severe as the number of tasks increases, and $ii$) difficult tasks are more likely to have conflicts with other tasks. Notably, MoORE outperforms all other baselines across all settings.

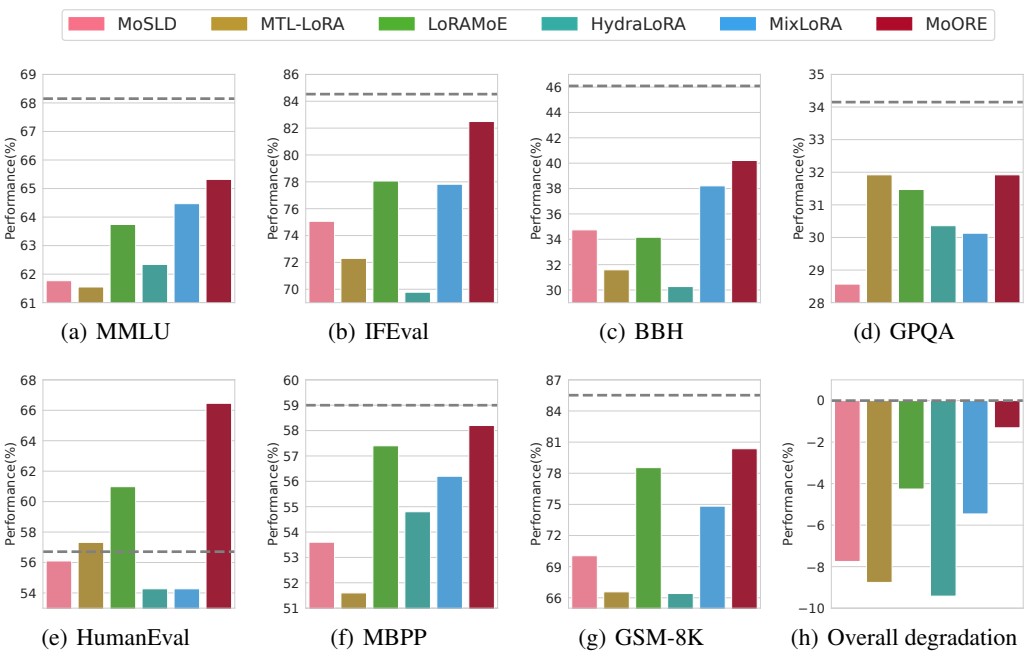

Figure 2: The loss of performance in OR-MTL. (a-g) The gray dashed line represents the performance of LLaMA-3.1 8B before adaptation. (h) The overall performance degradation across all tasks.

## 4.3 Performance in Oblivion-Resistance

To demonstrate the superiority of MoORE in mitigating task oblivion, we compare various methods in the Language Model Evaluation Harness framework [19]. In particular, we first adapt the base model on CSR-MTL using different methods. Then, we evaluate the adapted models on OR-MTL, comparing them with the base model. Figure 2 shows that MoORE consistently mitigates task oblivion across all seven tasks of OR-MTL, with an average performance drop of only 1.31% compared to the base model. It significantly outperforms the other adaptation methods. On HumanEval, MoORE achieves performance exceeding the original model, with an improvement of 9.75%. Similarly, LoRAMoE and MTL-LoRA also obtain slight improvements of 4.27% and 0.16%, respectively. This intriguing phenomenon may imply that the datasets in CSR-MTL have some relevance to HumanEval, providing information and capabilities beneficial for solving this task.

**Oblivion-resistance regarding task number and difficulty.** Furthermore, we conduct experiments to investigate how the ability of oblivion-resistance changes with increased task number and difficulty. The results in Figure 1(c) show that after adapting to the top-3 tasks of CSR-MTL, MoORE and its strongest competitor, LoRAMoE [14], achieve comparable performance on OR-MTL. However, with the increase of task number and difficulty, the performance of LoRAMoE changes slightly while the performance of MoORE increases and becomes superior to that of LoRAMoE. This result demonstrates that MoORE has stronger oblivion-resistance than its competitors.

**The impact of task correlation.** We investigate the reason for MoORE's oblivion-resistance empirically. In particular, we consider the samples of several sub-tasks in MMLU and those of six tasks in CSR-MTL. For each task/sub-task, we compute the average weights of MoORE's experts over its samples, i.e., $\boldsymbol{g}_k = \frac{1}{|\mathcal{D}_k|} \sum_{\boldsymbol{x}^{(k)} \in \mathcal{D}_k} g(\boldsymbol{x}^{(k)})$. Given a sub-task of MMLU and a task of CSR-MTL, denoted as $\mathcal{D}_k$ and $\mathcal{D}_{k'}$, respectively, we measure their correlation by $\|\boldsymbol{g}_k - \boldsymbol{g}_{k'}\|_2$. For each sub-task of MMLU, we record the performance degradation of MoORE compared to the base model. Figure 3 shows normalized performance degradation and task correlation. This visualization result indicates that the oblivion-resistance arises from the correlation of tasks — for correlated tasks, the model can learn some common domain knowledge during adaptation and thus avoid catastrophic forgetting.

**The generalization to unseen task IDs.** The computation of task-level weights requires task IDs as input, but the ID of a new task is unavailable during inference. Therefore, we use the average of

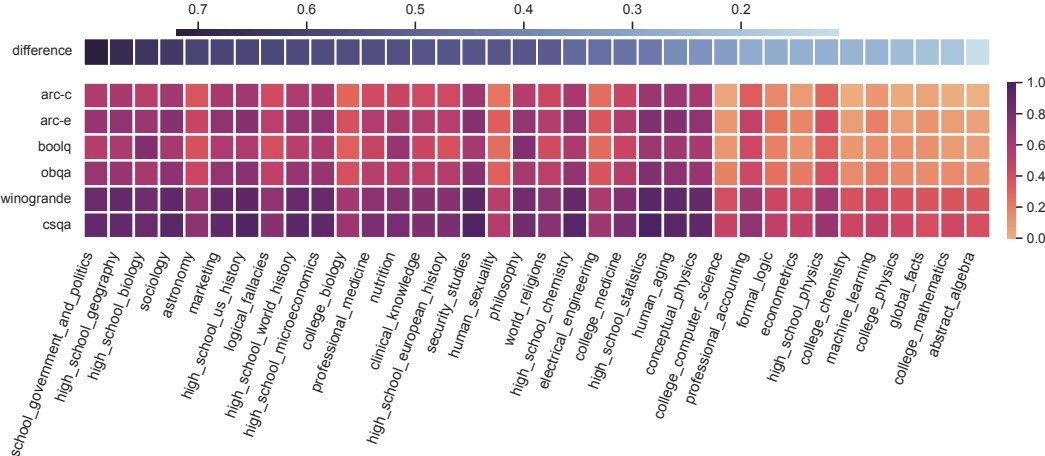

Figure 3: The visualization of normalized performance degradation and task correlation. The "difference" shown in the first row is the normalized performance degradation, i.e., $(\text{Acc}_{\text{Base}} - \text{Acc}_{\text{MoORE}})/100\%$. The following matrix records the normalized task correlation. The element in the $j$-th row and the $i$-th column is $\|\boldsymbol{g}_i - \boldsymbol{g}_j\|_2 / \max_{k,k'} \|\boldsymbol{g}_k - \boldsymbol{g}'_k\|_2$.

the existing task ID embeddings as the embedding of the new task. Specifically, after adapting the base model on CSR-MTL, MoORE only learns the task IDs of the nine tasks in CSR-MTL. During evaluation on OR-MTL, we use the average of the nine task ID embeddings as the task ID embedding of each new task. The results in Figure 1(c) and 2 validate the effectiveness of this approach.

In addition, this experiment also explains the result in Figure 1(c). In particular, the more tasks considered in the adaptation phase, the more likely some tasks correlate with those covered in the pre-training phase. As a result, increasing the number of tasks in the adaptation phase helps enhance the oblivion-resistance of MoORE.

### 4.4 Inference Efficiency

To compare inference efficiency, we adapt the base model using MoORE and other baselines, and evaluate their inference time on GPQA. For a fair comparison, all experiments are conducted on a single NVIDIA A100 GPU with the same batch size. The results in Figure 1(d) show the average inference time per batch for each method, which indicates that MoORE achieves superior inference efficiency compared to most baselines and is comparable to MixLoRA. Compared to the base model (i.e., the gray dashed line), MoORE moderately increases inference time.

## 5 Conclusion and Future Work

In this paper, we propose a simple but effective model MoE-ization strategy for multi-task adaptation, leading to a novel MoE architecture, called MoORE. Experiments on various datasets show the effectiveness and superiority of MoORE, demonstrating that imposing orthogonality on experts and maintaining the column space of the pre-trained weight matrix help improve the resistance of the adapted model to task conflict and oblivion.

**Limitations and future work.** As shown in Figure 1(d), the inference efficiency of MoORE is comparable to that of baselines. However, further reducing its complexity for large-scale applications is challenging. In particular, MoORE applies many more experts than classic MoE models, and the analytic experiments in Appendix C.4 show that it learns dense weights for the experts. In addition, besides imposing orthogonal adapters, we need to find a more effective solution to enhance model capacity in broader scenarios. In theory, the results in Figures 1(b) and 1(c) show that increasing the number of tasks leads to a higher risk of task conflict while helping mitigate task oblivion. How to understand this "trade-off"? How can the limitation in multi-task adaptation be quantified regarding the number of tasks? These are still open problems. Based on the above analysis, we plan to explore efficient implementations of MoORE and study its theoretical properties in the future.

## Acknowledgments and Disclosure of Funding

This work was supported by National Natural Science Foundation (92270110), Beijing Natural Science Foundation (L233008), the Fundamental Research Funds for the Central Universities, and the Research Funds of Renmin University of China. We also acknowledge the support provided by the fund for building world-class universities (disciplines) of Renmin University of China and by the funds from Engineering Research Center of Next-Generation Intelligent Search and Recommendation, Ministry of Education, and from Intelligent Social Governance Interdisciplinary Platform, Major Innovation & Planning Interdisciplinary Platform for the "Double-First Class" Initiative, Renmin University of China.

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

# A   Hyperparameters and Implementation Details

The detailed hyperparameter setups are presented in Table 7. Both training and testing are conducted on one NVIDIA A100 GPU.

Table 7: Hyperparameter configurations of MoORA on the CSR-MTL and the NLU-MTL.

| Hyperparameters | MoORA |
|---|---|
| Cutoff Length | 512 |
| Batch Size | 8 / 64 |
| Epochs | 2 / 5 |
| Learning Rate | 3E-04 |
| LR scheduler | Warmup-Stable-Decay |
| Warmup Ratio | 5% |
| Decay Ratio | 5% |
| Optimizer | AdamW |
| Dropout Rate | 0.0 |
| Target Modules | Q, K, V, O, Up, Down, Gate |
| $D_t$ | 128 |
| $D_s$ | 64 |
| $L$ | 8 |

# B   Tasks and Datasets

Detailed information about the CSR-MTL, the NLU-MTL and the OR-MTL is presented in Tables 8, Table 9 and Table 10, respectively. These tables include the sizes of the training and test sets, as well as the task types.

Table 8: The basic information of CSR-MTL.

| Task Name | #Train | #Test | Task Type |
|---|---|---|---|
| ARC-Challenge | 1,119 | 1,172 | Question Answering |
| ARC-Easy | 2,250 | 2,380 | Question Answering |
| BoolQ | 9,427 | 3,270 | Text Classification |
| OpenBookQA | 4,957 | 500 | Question Answering |
| PIQA | 16,100 | 1,840 | Question Answering |
| SocialIQA | 33,410 | 1,954 | Question Answering |
| HellaSwag | 39,905 | 10,042 | Sentence Completion |
| WinoGrande | 9,248 | 1,267 | Fill in the Blank |
| CommonsenseQA | 9,741 | 1,140 | Question Answering |

Table 9: The basic information of NLU-MTL.

| Task Name | #Train | #Validation | Task Type |
|---|---|---|---|
| CoLA | 8,551 | 1,043 | Text Classification |
| MNLI | 392,702 | 9,815 | - |
| MRPC | 3,668 | 408 | - |
| QNLI | 104,743 | 5,463 | - |
| QQP | 363,846 | 40,430 | - |
| RTE | 67,350 | 873 | - |
| SST-2 | 2,491 | 277 | - |

Table 10: The basic information of OR-MTL.

| Task Name | #Test | Task Type |
|---|---|---|
| MMLU | 14,042 | Question Answering |
| IFEval | 541 | Text Generation |
| BBH | 6,511 | Question Answering |
| GPQA | 448 | Question Answering |
| HumanEval | 164 | Text Generation |
| MBPP | 500 | Text Generation |
| GSM-8K | 1,319 | Text Generation |

# C  More Experimental Results

## C.1  Performance on other Model Architectures in Conflict-Resistance

To evaluate the cross-model performance of MoORE, we take Qwen-2.5 7B as the backbone model and apply various MoE-based multi-task adaptation methods to it. Experimental results on CSR-MTL are shown in Table 11. We can find that MoORE outperforms baselines consistently under different hyperparameter settings, and the performance of MoORE is robust to the change of $L$. These results verify the robustness of our method and its compatibility with various model architectures.

Table 11: Results (%) of Qwen-2.5 7B adapted by various methods on CSR-MTL. The best results on each dataset are shown in **bold**, and the second best results are shown in underline.

| Method | #Params | ARC-C | ARC-E | BoolQ | OBQA | PIQA | SIQA | HellaS | WinoG | CSQA | Overall |
|---|---|---|---|---|---|---|---|---|---|---|---|
| LoRA [25] | 2.12% | 87.29 | 91.88 | 70.37 | 90.60 | 87.92 | 79.02 | 92.73 | 74.98 | 84.36 | 84.35 |
| MixLoRA [30] | 3.32% | 86.86 | 91.75 | 71.38 | 91.40 | 89.28 | 80.45 | 95.06 | 78.45 | 85.09 | 85.53 |
| MoSLD [81] | 1.43% | 86.69 | 92.13 | 71.96 | 90.40 | 89.66 | 80.40 | 95.17 | 78.30 | **85.50** | 85.58 |
| HydraLoRA [60] | 2.83% | 88.23 | 91.54 | 72.57 | 91.20 | 85.80 | 80.09 | 95.95 | **82.24** | 85.26 | 85.87 |
| MTL-LoRA [73] | 2.80% | 87.80 | 92.09 | 72.72 | 92.20 | 86.07 | 79.79 | 95.92 | 80.66 | 84.36 | 85.73 |
| LoRAMoE [14] | 2.21% | 87.20 | 91.84 | **73.61** | 90.40 | **90.37** | **81.27** | **95.99** | 80.27 | 84.60 | 86.17 |
| MoORE $_{L=0}$ | 2.29% | 88.06 | 92.17 | 73.49 | 92.20 | **90.37** | 79.58 | 95.40 | 81.53 | 84.03 | 86.31 |
| MoORE $_{L=2}$ | 2.32% | 88.23 | 92.34 | 73.09 | 91.60 | 90.32 | 80.60 | 95.50 | 81.45 | 84.19 | 86.37 |
| MoORE $_{L=4}$ | 2.35% | **88.40** | 92.21 | 73.18 | 91.40 | 90.21 | **81.27** | 95.25 | 80.58 | 84.93 | 86.38 |
| MoORE $_{L=16}$ | 2.53% | 88.23 | **92.59** | 73.30 | **92.40** | 90.26 | 81.12 | 95.50 | 81.37 | 84.93 | **86.63** |

## C.2  Numerical Performance in Oblivion-Resistance

Following LLaMA-3.1 8B, we select MMLU [24, 23], IFEval [83], BIG-Bench Hard (BBH) [58], GPQA [51], HumanEval [7], MBPP [4], and GSM-8K [10] as evaluation tasks and adopt the same few-shot settings. More detailed results of the oblivion-resistance experiments are presented in Table 12, corresponding to the results shown in Figures 1(c) and 2.

## C.3  Ablation Study

To analyze the impact of the task-level routing, sample-level routing, and orthogonal adapter quantitatively, we conduct ablation study on the CSR-MTL dataset. In Table 13, we record the overall performance achieved when only one of these three modules is applied. In particular, without the orthogonal adapter, our method degrades to the MoE with fixed experts. When merely applying the orthogonal adapter without any routing, our method is simplified as the single-task adaptation method HRA [75]. The results show that when only one of these three modules is applied, even with a wider network and more trainable parameters, the model's performance declines significantly across the entire dataset. These results demonstrate that all these three modules contribute to the overall effectiveness of MoORE.

Table 12: The loss of performance in the original tasks. Each method is represented by two rows of data: the first row indicates the performance on the current task, while the second row shows the difference between the post-fine-tuning score and the pre-fine-tuning score.

| Category | General | | | Reasoning | Code | | Math | |
|---|---|---|---|---|---|---|---|---|
| Task | MMLU | IFEval | BBH | GPQA | HumanEval | MBPP | GSM-8K | Overall |
| #Shots | 5 | | 0 | 0 | 0 | 3 | 8 | |
| Llama-3.1-8B-Instruct | 68.15 | 84.53 | 46.09 | 34.15 | 56.71 | 59.00 | 85.52 | 62.02 |
| LoRA | 32.40 | 29.14 | 13.65 | 27.23 | 0.00 | 0.00 | 2.12 | 14.93 |
| | -35.75 | -55.39 | -32.44 | -6.92 | -56.71 | -59.00 | -83.40 | -47.09 |
| LoRAMoE | 63.74 | 78.06 | 34.16 | 31.47 | 60.98 | 57.40 | 78.54 | 57.76 |
| | -4.41 | -6.47 | -11.93 | -2.68 | 4.27 | -1.60 | -6.98 | -4.26 |
| MoSLD | 61.77 | 75.06 | 34.74 | 28.57 | 56.10 | 53.60 | 70.05 | 54.27 |
| | -6.38 | -9.47 | -11.35 | -5.58 | -0.61 | -5.40 | -15.47 | -7.75 |
| MTL-LoRA | 61.55 | 72.30 | 31.59 | 31.92 | 57.32 | 51.60 | 66.57 | 53.26 |
| | -6.60 | -12.23 | -14.50 | -2.23 | 0.61 | -7.40 | -18.95 | -8.76 |
| HydraLoRA | 62.34 | 69.78 | 30.27 | 30.36 | 54.27 | 54.80 | 66.41 | 52.60 |
| | -5.81 | -14.75 | -15.82 | -3.79 | -2.44 | -4.20 | -19.11 | -9.42 |
| MixLoRA | 64.47 | 77.82 | 38.21 | 30.13 | 54.27 | 56.20 | 74.83 | 56.56 |
| | -3.68 | -6.71 | -7.88 | -4.02 | -2.44 | -2.80 | -10.69 | -5.46 |
| MoORE | 65.32 | 82.49 | 40.21 | 31.92 | 66.46 | 58.20 | 80.36 | 60.71 |
| | -2.83 | -2.04 | -5.88 | -2.23 | 9.75 | -0.80 | -5.16 | -1.31 |

Table 13: Ablation studies of MoORE on CSR-MTL. ✖ indicates the absence of that component.

| Method | $D_t$ | $D_s$ | $L$ | #Params | Overall |
|---|---|---|---|---|---|
| Merely Apply Task-level Routing | 16 | ✖ | ✖ | 0.14% | 79.55 |
| | 32 | ✖ | ✖ | 0.29% | 81.80 |
| | 64 | ✖ | ✖ | 0.58% | 83.06 |
| | 128 | ✖ | ✖ | 1.16% | 83.53 |
| Merely Apply Sample-level Routing | ✖ | 16 | ✖ | 0.39% | 82.30 |
| | ✖ | 32 | ✖ | 0.78% | 82.91 |
| | ✖ | 64 | ✖ | 1.57% | 83.57 |
| | ✖ | 128 | ✖ | 3.13% | 83.80 |
| Merely Apply Orthogonal Adapter | ✖ | ✖ | 2 | 0.03% | 80.76 |
| | ✖ | ✖ | 4 | 0.06% | 82.15 |
| | ✖ | ✖ | 8 | 0.12% | 82.98 |
| | ✖ | ✖ | 16 | 0.25% | 83.90 |
| MoORE | 128 | 64 | 8 | 2.84% | 85.11 |

## C.4 Analysis of Routing Weights

To investigate the router's preferences for the experts in MoORE, we analyze the distribution of routing weights across different tasks in the CSR-MTL dataset. Take the $Q$ module from the first attention layer of the adapted model as an example. For each task, we compute and show the mean (red point) and variance (blue stem) of each expert's weight over all samples of the task in Figures 4 and 6. In each figure, the mean weight of an expert is shown as a red point, and the blue stem associated with the point shows the standard deviation. These visualization results provide us with some insights in the mechanism behind MoORE:

- **Preference on the originally important experts.** Compared to other components, the routing weights corresponding to the principal components of $W$ (i.e., the experts corresponding to large singular values) are adjusted more significantly. The routing weights corresponding to the significant experts are distributed relatively symmetrically around the zero scale, whereas the routing weights for the minor components are more concentrated above the zero scale. This

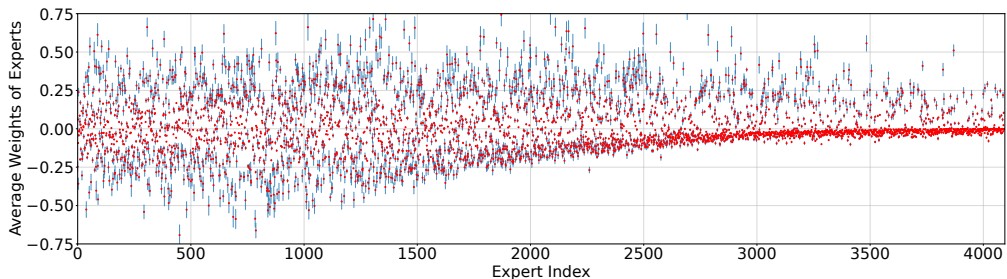

Figure 4: The mean and variance of routing weights obtained in HellaS.

indicates that the model tends to focus on adjusting the routing weights of the experts that are important in the pre-training phase. In other words, these originally important experts often maintained their significance when adapting for new task.

- Based on the mean and variance of the routing weights, the nine tasks in CSR-MTL can be grouped into three categories:

    – **Large Mean, Large Variance:** Tasks such as Hellaswag fall into this group. MoORE performs well on the Hellaswag. A large mean indicates that the model significantly enhances or discards well-learned pretraining knowledge, while a large variance means that the model can capture substantial differences between samples and assign different experts to them with high flexibility.

    – **Small Mean, Large Variance:** Tasks such as OBQA, PIQA, SIQA, and CSQA belong to this group. This result indicates that for each of these tasks, its samples have significant diversity, and MoORE needs to activate different experts to handle different samples.

    – **Small Mean and Variance:** Tasks such as ARC-C, ARC-E, BoolQ, and Winogrande (WinoG) are in this group. Except for ARC-E, MoORE performs poorly on these tasks. A small mean indicates minimal adjustment to pretraining knowledge, and a small variance suggests minor differences between samples. For the relatively simple ARC-E task, the model may have already mastered the ability to solve it, requiring little adjustment. However, for the more challenging tasks in this group, the model may struggle to learn the ability to solve them within the column space spanned by the singular vectors.

## D  Broader Impacts

MoORE can effectively enhance the performance of LLMs in multi-task adaptation and facilitate the application of LLMs in real-world scenarios. Similar to other multi-task adaptation methods, such as MixLoRA and HydraLoRA, MoORE may result in some potential negative social impacts, such as the risk of abuse for adapting LLMs in illegal activities. However, these issues are not unique to MoORE — other adaptation methods are also susceptible to similar risks. Addressing these issues thoroughly will require advancements in technology, the establishment of relevant legal frameworks, and shifts in societal perspectives.

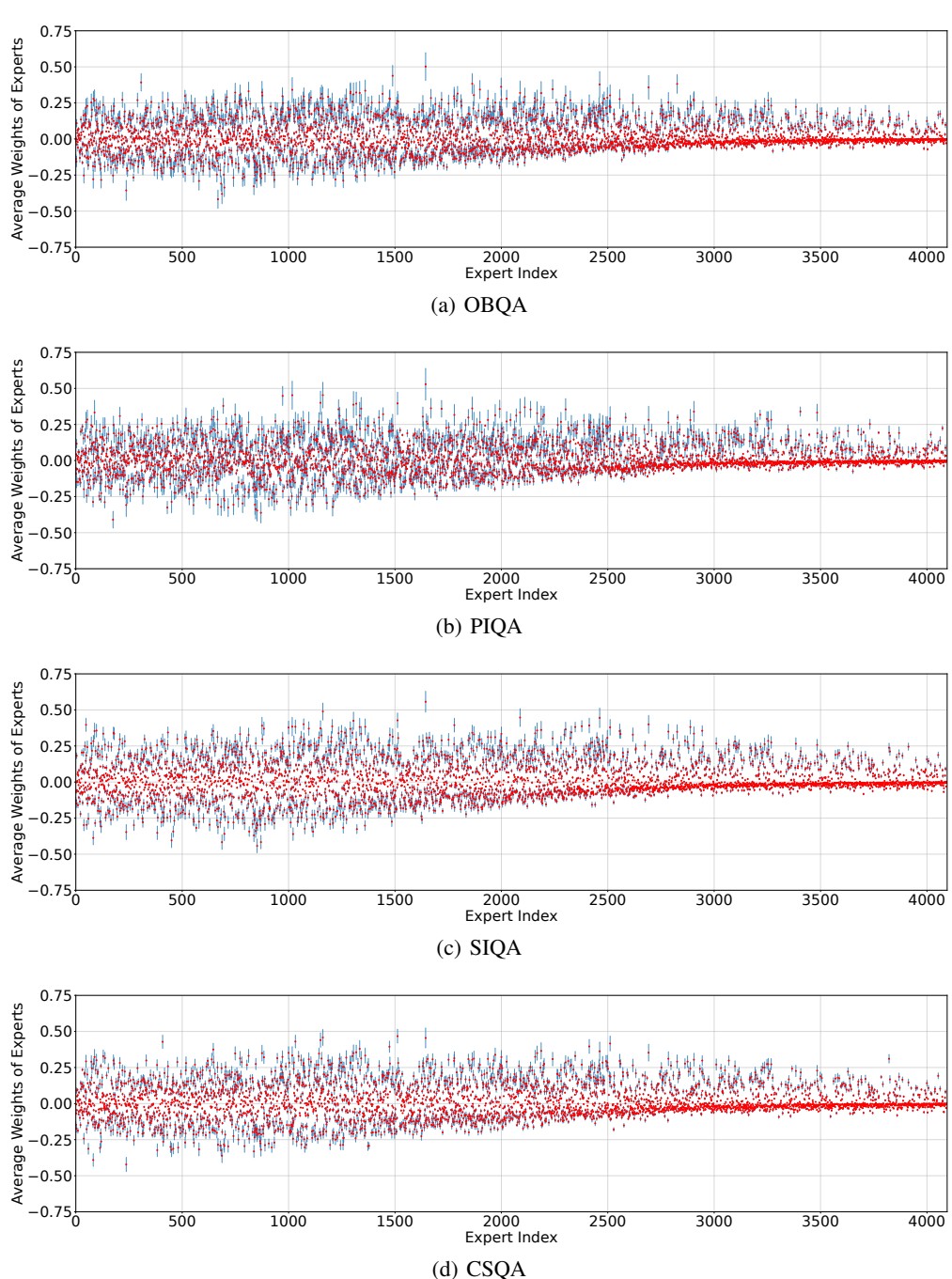

Figure 5: The mean and variance of routing weights obtained in four QA tasks.

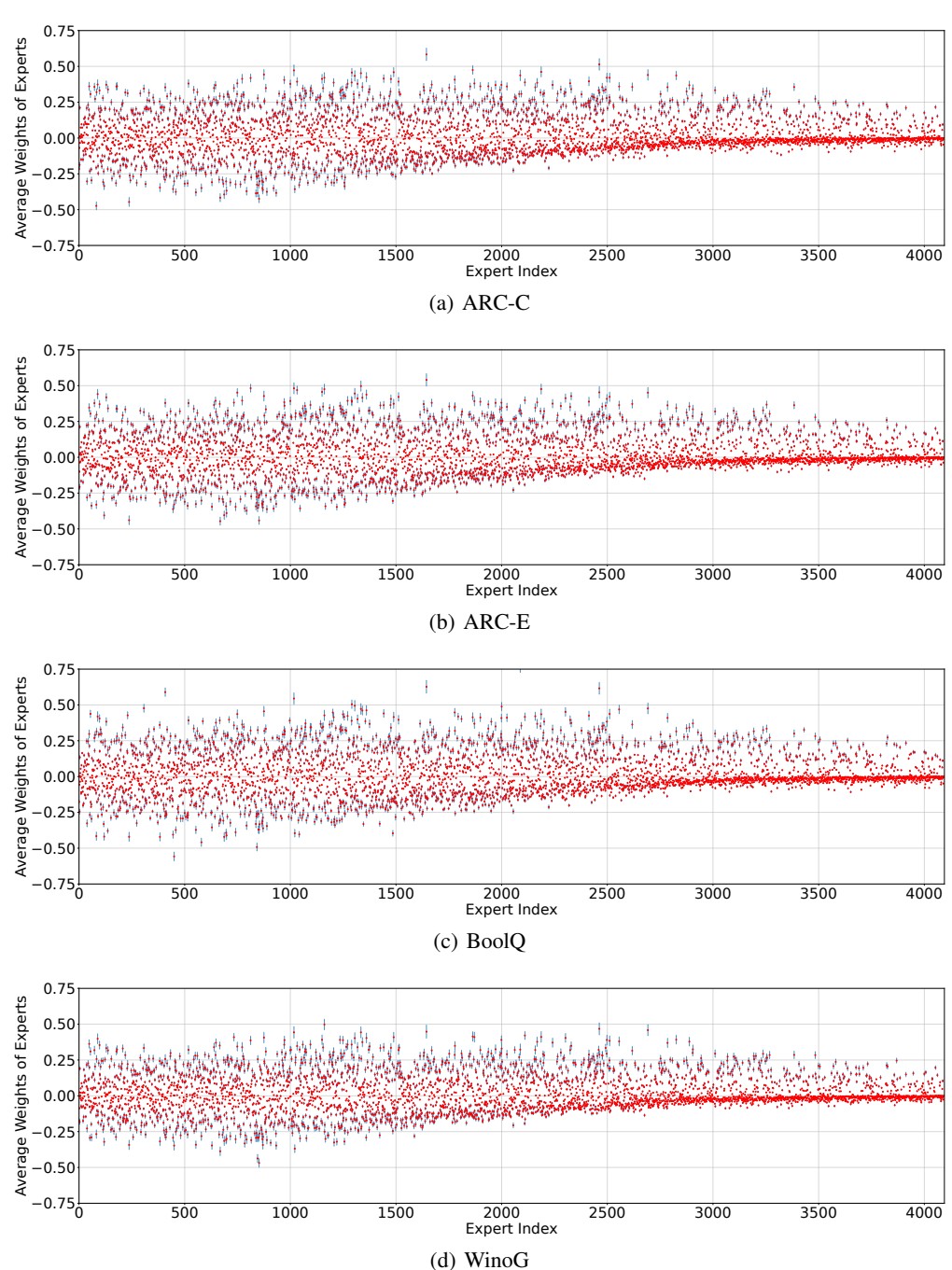

Figure 6: The mean and variance of routing weights obtained in the remaining four tasks of CSR-MTL.

