# OpenReview forum: "MoORE: SVD-based Model MoE-ization for Conflict- and Oblivion-Resistant Multi-Task Adaptation"
_NeurIPS.cc/2025/Conference — NeurIPS 2025 poster_

### Official Review · Reviewer_5xhJ · 2025-06-20

**Clarity:** 3
**Significance:** 3
**Originality:** 2
**Rating:** 4
**Confidence:** 4

**Summary:**

This paper addresses the challenges of task conflict and forgetting in multi-task adaptation of large-scale foundation models by proposing MoORE. MoORE applies SVD to the pre-trained weights and introduces a learnable router to dynamically combine orthogonal rank-one experts. To ensure fair comparison, all the compared methods are built upon the same base model, and extensive experiments across multiple datasets are conducted to verify the effectiveness of MoORE.

**Questions:**

1. I'm a bit confused about the title — why is there a hyphen between “Conflict” and “and”, but no hyphen between “and” and “Oblivion”? That seems a bit odd.

2. The manuscript claims to validate the superiority of the proposed method "MoORE" in mitigating task oblivion. However, Figure 2 presents results for "SVDMoE" rather than "MoORE," leading to potential confusion about whether they are equivalent. The authors should clarify the relationship between MoORE and SVDMoE and use consistent terminology throughout the paper.

3. The abstract claims MoORE demonstrates superiority in terms of efficiency. Could the authors clarify what aspects “efficiency” encompasses—is it a composite metric or specific factors? Also, please specify where in the paper this efficiency is demonstrated experimentally or analytically. Clearer definitions and supporting evidence are recommended.

**Ethical Concerns:**

["NO or VERY MINOR ethics concerns only"]

**Final Justification:**

The authors have clarified my comments, so I keep my positive rating.

**Limitations:**

Yes.

**Paper Formatting Concerns:**

N/A.

**Quality:**

3

**Strengths And Weaknesses:**

Strengths:

1.The paper tackles an important problem in multi-task learning with large-scale foundation models.

2.The paper conducts extensive comparisons and evaluations on diverse datasets to verify the effectiveness of the proposed method.

Weaknesses：

1.The proposed method in the paper does not achieve the best performance on all datasets.

2.The paper claims superiority in efficiency in the abstract, but later notes that MoORE increases inference time compared to the original LLaMA-3.1 8B. This seems inconsistent.

---

> ### Author Rebuttal · Authors · 2025-07-30
>
> Thank you for your appreciation of our work. We provide the responses below to resolve your concerns one by one.
>
> **Q1: The proposed method in the paper does not achieve the best performance on all datasets.**
>
> **A1:** Instead of fine-tuning $K$ different models for $K$ tasks, multi-task adaptation aims to learn a single model to deal with $K$ tasks. **In multi-task adaptation scenarios, the average model performance over all tasks is the most critical evaluation metric, which reflects the resistance of the model to task conflict and oblivion.** In particular, given the datasets of $K$ tasks, the target model should obtain strong overall performance across all tasks, rather than performing strongly in some tasks while degrading severely in the other ones.
>
> Based on the above information, although MoORE does not achieve the best performance on all datasets, we have demonstrated its superiority in terms of overall performance and provided the following evidence in our submission.
>
> **1) Superiority on mitigating task conflict:** In Figure 1(b), the "Overall" columns in Tables 3 and 4, MoORE outperforms its competitors consistently in various multi-task adaptation scenarios, which means that it mitigates the conflicts across different tasks, achieving the best average performance over all the tasks.
>
> **2) Superiority on mitigating task oblivion:** In Figure 1(c), Figure 2, and Table 9 (in Appendix), MoORE mitigates the loss of model capability achieved in the pre-trained phase. In particular, when adapting LLaMA using the baseline methods, the adapted models tend to forget the capability of the original LLaMA, suffering severe performance degradation in the original tasks that LLaMA performs well. Our MoORE can mitigate this phenomenon significantly --- **in the original tasks that LLaMA performs well, the model obtained by MoORE achieves comparable overall performance.**
>
> **Q2: The definition and evidence for the efficiency of MoORE**
>
> **A2:** Thank you for your question. Due to space limitations, our explanation of "efficiency" in the paper was somewhat brief, and we will improve this aspect in the revision. The detailed explanation is shown below for a quick review.
>
> **In our paper, "efficiency" refers specifically to the inference time of MoE-based multi-task adaptation.**
>
> 1) The results in Figure 1(d) show that MoORE is comparable to MixLoRA in inference time, but outperforms all other baselines.
>
> 2) The results in Figure 1(b) show that MixLoRA significantly sacrifices model performance when pursuing computational efficiency, while MoORE maintains high performance with comparable complexity.
>
> These results provide strong evidence demonstrating MoORE’s superiority in terms of efficiency compared to other MoE-based multi-task adaptation methods.
>
> Note that, because of the router in MoE architectures, all the MoE-based multi-task adaptation methods (including MoORE and its competitors) introduced additional trainable parameters that cannot be fully merged into the original parameter matrix.
> Therefore, these methods introduce additional computations, making the increase in inference time unavoidable compared to the original LLaMA model. **Therefore, the "superiority in efficiency" claimed in the abstract refers to MoORE’s performance compared to other MoE-based baselines rather than the original LLaMA model.**
>
>
> **Q3: The grammar correctness of the hyphen between “Conflict” and “and”**
>
> **A3:** The complete expression should be “Conflict-Resistant and Oblivion-Resistant”.  This hyphenation (“Conflict- and Oblivion-Resistant”) is a convention to avoid repeating the word “Resistant”, which is commonly used in English.
>
>
> **Q4: The inconsistency of the method name --- MoORE or SVDMoE**
>
> **A4:** Thank you for pointing out the mistake. The "SVDMoE" in Figure 2 should be "MoORE", and we will correct this in the revision.
>
> In summary, we hope the responses above can resolve your concerns successfully and increase your confidence in raising the rating score of our submission. Feel free to contact us if you have additional comments in the following discussion phase.

---

> ### Comment · Reviewer_5xhJ · 2025-08-03
>
> The authors have clarified my comments. Since my rating has been already positive, I would like to maintain my rating.

---

> > ### Author Response · Authors · 2025-08-03
> >
> > Thanks for your appreciation of our work. If you have any further comments or questions, we would be happy to address them.

---

### Official Review · Reviewer_KfX6 · 2025-06-27

**Clarity:** 3
**Significance:** 2
**Originality:** 3
**Rating:** 4
**Confidence:** 2

**Summary:**

This paper proposes converting the dense pre-trained model weights into low-rank expert models and dynamically combining these expert models to learn new tasks as much as possible while reducing the forgetting of general knowledge (or old knowledge).

**Questions:**

In SVD decomposition, directions with larger singular values are more important. Does this mean that most samples/tasks will choose the rank-one experts associated with the larger singular values, thus exacerbating the expert routing imbalance?

**Ethical Concerns:**

["NO or VERY MINOR ethics concerns only"]

**Final Justification:**

I am not particularly familiar with the content of this work, so I tend to give positive scores and listen to the opinions of other experts.

**Limitations:**

yes

**Quality:**

3

**Strengths And Weaknesses:**

**Strengths:**
- This paper proposes splitting a dense pre-trained model into a series of rank-one experts and dynamically combining these experts through a gating network.
- The proposed method has been validated on a large number of commonsense reasoning (CSR) and natural language understanding (NLU) tasks.

**Weaknesses:**
- The meaning of adding orthogonal transformations to the input is unclear. Simply interpreting it as increasing representational capacity is insufficient, and the results in Table 4 also show that increasing the orthogonal projection of the input is not always effective.
- In the last column of Figure 2, it should be MoORE instead of SVDMoE? According to the results in Figure 2, the proposed MoORE still shows a significant performance drop on general tasks (even though relative to other methods, the drop is smaller), such as a 6% drop on the BBH task and about a 3% drop on MMLU.
- It is unclear how the proposed method compares with full fine-tuning (upper bound) on CSR and NLU datasets in terms of performance differences.
- Existing works have also discussed applying orthogonal constraints to LoRA [1], which needs to be discussed.
- Existing works have also explored splitting dense models into MoE models [2,3], and these works need to be discussed and compared.

[1]Wang, Xiao, et al. "Orthogonal subspace learning for language model continual learning." EMNLP  (2023).

[2]Komatsuzaki, Aran, et al. "Sparse upcycling: Training mixture-of-experts from dense checkpoints." ICLR (2023).

[3]Wu, Haoyuan, et al. "Parameter-Efficient Sparsity Crafting from Dense to Mixture-of-Experts for Instruction Tuning on General Tasks." Proceedings of the 2024 Conference on Empirical Methods in Natural Language Processing. 2024.

---

> ### Author Rebuttal · Authors · 2025-07-31
>
> Thanks for your appreciation of our work and constructive comments. Below, we try to resolve your concerns one by one.
>
>
> **Q1: The reason for applying orthogonal transformations**
>
> **A1:** Firstly, as stated in Line 161, we aim to maintain the orthogonality among experts during training because this can reduce redundant knowledge and help mitigate task conflict. In MoORE, each expert is constructed by the outer product of singular vectors. Updating the experts directly by stochastic gradient descent may lead to the loss of orthogonality. Therefore, we introduce $L$ learnable Householder reflections (i.e., the proposed orthogonal transformations) into the experts, making them learnable and maintaining the orthogonality throughout the training process.
>
> In Lines 235-242, we have analyzed the results of NLU-MTL shown in Table 4. In particular, the natural language understanding tasks in NLU-MTL mainly rely on the pretraining knowledge, so adding additional learnable parameters is redundant to some degree. However, in the CSR-MTL experiment, applying the learnable orthogonal transformations helps improve model performance consistently, as shown in Table 3.
>
>
> **Q2: The inconsistency of the method name, and the loss of performance in OR-MTL**
>
> **A2:** Firstly, thank you for pointing out the inconsistency of our method name. The last column in Figure 2 should be MoORE, and we will correct this in the revision.
>
> Secondly, the results in Figure 2 are not in conflict with our statement in Line 173. In particular, we claim that MoORE can **"mitigate"** task oblivion and achieve better multi-task adaptation results than existing solutions. It does not mean that our method can completely **"eliminate"** this issue. We admit that MoORE may still exhibit some degree of task oblivion after being trained on other tasks. However, the results in Figure 1(c), Figure 2, and Table 9 have demonstrated that MoORE has a stronger ability to resist task oblivion compared to the baselines. In summary, MoORE indeed outperforms the baselines consistently in various multi-task adaptation scenarios, showing better resistance to task conflict and oblivion.
>
>
> **Q3: The comparison with full fine-tuning.**
>
> **A3:** Firstly, we need to emphasize that in multi-task adaptation scenarios, full fine-tuning does **NOT** achieve the upper bound performance compared to the MoE-based methods (including MoORE and the MoE-based baselines).
>
> In particular, fine-tuning the whole pre-training model on the data of $K$ tasks can outperform LoRA because it adjusts all model parameters rather than a low-rank adapter. However, full fine-tuning does not change the model architecture and does not introduce additional parameters or modules. In contrast, MoE-based methods introduce additional modules, e.g., the baselines introduce routers and external low-rank experts to the model, and our MoORE introduces a router and a learnable orthogonal transformation. As a result, the MoE-based methods often work better than full fine-tuning in multi-task adaptation scenarios.
>
> Following your suggestion, we compare full fine-tuning with the other methods in adapting LLaMA-3.1 8B on CSR. The results shown below verify our claim - full fine-tuning works better than LoRA but can be inferior to state-of-the-art MoE-based methods, e.g., LoRAMoE and our MoORE.
>
> |Method|CSR-MTL|
> |-|-|
> |Full Fine-tuning|84.27|
> |LoRA|79.54|
> |MixLoRA|82.55|
> |MoSLD|83.55|
> |HydraLoRA|83.84|
> |MTL-LoRA|84.09|
> |LoRAMoE|84.34|
> |MoORE L=0|84.98|
> |MoORE L=2|85.02|
> |MoORE L=4|85.02|
> |MoORE L=8|85.11|
>
>
> **Q4: Add a comparison with the LoRA method applying an orthogonal constraint, e.g., the O-LoRA in the reference [1] mentioned in the comment.**
>
> **A4:** Thank you for your suggestion. In the revised paper, we will add O-LoRA [1] to the Related Work section and discuss the similarities and differences between O-LoRA and MoORE accordingly. However, it should be noted that the O-LoRA and MoORE have the following differences:
>
> 1. MoORE constructs strictly orthogonal experts based on SVD, while O-LoRA uses a regularization term to pursue orthogonality, which can not strictly make its LoRA adapters orthogonal to each other.
>
> 2. O-LoRA does not adopt a MoE architecture. To deal with different tasks, O-LoRA adds new LoRA adapters in a continual learning scenario, rather than designing and learning a flexible routing mechanism for given and fixed experts. As a result, the model complexity of O-LoRA is linear to the number of tasks. From this viewpoint, the scalability of our MoORE is better.
>
>
> **Q5: Add a comparison with the references [2, 3] mentioned in the comment**
>
> **A5:** The work in [2] focuses on upcycling, i.e., initializing a sparsely activated MoE model from a dense model checkpoint to reduce the pre-training cost, rather than adapting a pre-trained dense model in multi-task learning scenarios. The scope of this work [2] does not overlap with ours, and the method cannot be applied directly to our task.
>
> In our experiments, we find the performance of the work in [3] is similar to that of MixLoRA. Essentially, this is because this work is a simplified version of MixLoRA:
>
> 1. Both of them perform MoE modifications in the FFN layer. For this method, each expert in an MoE module is a LoRA adapter, and the experts share the same FFN parameters.
>
> 2. For the expert in the work [3], a LoRA with a residual connection is applied in a "sequential" manner. On the contrary, MixLoRA has more learnable parameters, and the width of each layer is larger than [3]. In particular, MixLoRA inserts additional LoRA adapters into attention layers, and within each expert, the FFN and LoRA are in a "parallel" structure.
>
> We will add the work in [2, 3] to the Related Work section and discuss them accordingly.
>
>
> **Q6: The risk of expert routing imbalance**
>
> **A6:** The expert routing imbalance issue (or called load imbalance issue) only arises in sparsely activated MoE architectures (such as MixLoRA). In contrast, our proposed MoORE activates all experts (this design is also adopted by LoRAMoE, HydraLoRA, and MTL-LoRA), so there is no load imbalance problem.
>
>
> In summary, we hope our responses can resolve your concerns successfully and increase your confidence in raising your rating score. It would be appreciated if you could further support our work in the following decision phase.

---

> > ### Comment · Reviewer_KfX6 · 2025-08-06
> >
> > Thank you to the authors for your careful reply. Since I am not an expert in this field and our rating is already positive, I will keep this score.

---

> ### Comment · Area_Chair_tkVf · 2025-08-04
> **Author-Reviewer Discussion**
>
> Dear Reviewer KfX6,
>
> Thanks for taking the time to review for NeurIPS. Since we have fewer than three days before the author-reviewer discussion period ends (August 6, 11:59pm AoE), could you please do the following ASAP: carefully read the other reviews and the authors responses, and reply to the authors regarding whether your questions or concerns have been addressed. If any issues remain unresolved, please clearly specify them so that the authors have a fair opportunity to respond before the author-reviewer window closes.
>
> Thank you again for your contribution to the review process.
>
> Best,
> AC

---

### Official Review · Reviewer_eS7y · 2025-06-27

**Clarity:** 4
**Significance:** 2
**Originality:** 1
**Rating:** 2
**Confidence:** 4

**Summary:**

This paper presents MoORE, which repackages task-aware Singular Value Fine-tuning within a MoE story.

The authors decompose model weights via SVD, term each pair of left/right singular vectors as "rank-1 experts", and refer to the natural summation in SVD reconstruction as a "mixture" of experts. Projected singular values are rebranded as "routing weights" (router).

Methodologically, model weights first undergo SVD decomposition. Subsequently, a task_id is input to train two projections: one on singular values and another between inputs and model weights. Household adaptors are utilized to ensure orthogonality.

The core contribution lies in applying Singular Value Fine-tuning to address key multi-task adaptation challenges: task conflicts and catastrophic forgetting.

Theoretical underpinnings regarding orthogonality and column space preservation provide justification for the approach's efficacy.

Empirical results demonstrate modest improvements.

**Questions:**

- The paper's central methodology appears to be functionally equivalent to established Singular Value Decomposition (SVD) based fine-tuning techniques. Could the authors precisely articulate the fundamental technical novelty of "MoORE" that distinguishes it from prior work on using SVD for model adaptation? Specifically, what aspects of the proposed method constitute a genuine innovation beyond a rebranding of existing concepts?
- The paper frames its approach as "Model MoE-ization," yet the theoretical motivation and empirical evaluation appear disconnected from the established principles of Mixture-of-Experts (MoE) architectures. Given that the method operates on pre-trained models via SVD and traditional MoEs focus on sparse activation during training, what is the theoretical justification for this MoE analogy? Why decomposing pre-trained model weights should be considered "MoE-ization" beyond superficial structural similarities? Consequently, why were direct comparisons with genuine MoE models omitted from the experiments, which instead focused on LoRA-based methods?
- How does the proposed framework address scenarios where task identifiers are unavailable, such as in continual learning with unseen tasks or contexts where tasks are defined implicitly? This dependency seems to severely constrain the method's practical applicability in real-world settings.

**Ethical Concerns:**

["NO or VERY MINOR ethics concerns only"]

**Limitations:**

the authors adequately addressed the limitations and potential negative societal impact of their work

**Quality:**

2

**Strengths And Weaknesses:**

**Strengths:**
- Engineering-Friendly: The proposed method is straightforward to implement, requiring only SVD decomposition and lightweight projection layers, making it practically accessible for deployment.
- The paper exhibits clear exposition with logical flow, making the story easy to follow despite the conceptual reframing.

**Weaknesses:**
- Fundamental Lack of Novelty and Conceptual Misrepresentation: The core contribution appears to be a rebranding of established SVD fine-tuning techniques rather than genuine innovation. The authors introduce "Model MoE-ization" as a novel concept, but the proposed method shows no essential difference from existing singular value fine-tuning approaches ( e.g. https://proceedings.neurips.cc/paper_files/paper/2022/file/f3bfbd65743e60c685a3845bd61ce15f-Paper-Conference.pdf ). The MoE terminology creates conceptual confusion without adding substantive technical value. This MoORE is fundamentally singular value adaptation disguised with MoE nomenclature.
- Questionable Theoretical Motivation for "Model MoE-ization" and Misleading Framing: The rationale behind applying MoE-ization to already pre-trained language models remains unclear. Traditional MoE architectures primarily sparsify model parameters when processing vast training data to prevent overfitting, while the proposed SVD decomposition serves entirely different purposes. What meaningful purpose does "MoE-izing" an already-trained language model serve beyond SVD decomposition with alternative terminology? The authors fail to articulate why decomposing pre-trained model weights should be considered "MoE-ization" beyond superficial structural similarities.
- Despite the MoE framing, the experimental evaluation completely omits comparisons with actual MoE methods, instead focusing on LoRA-based approaches. This fundamental disconnect between the conceptual narrative and empirical validation raises serious concerns about the paper's positioning and undermines the paper's central narrative.
- Limited Generalizability due to Task_id Dependency: The method's reliance on explicit task_ids restricts its applicability to scenarios with tasks without a task_id. This fundamental limitation contradicts claims of general-purpose multi-task adaptation, as the approach cannot handle scenarios where task_id is unavailable or unseen.
- Inconsistent Performance Claims: The authors claim MoORE "consistently outperforms existing multi-task adaptation methods", yet the experimental results reveal performance dependence on hyperparameter L selection. The paper lacks comprehensive hyperparameter sensitivity analysis.
- Limited Model Scope: Despite that the MoORE theory seems to be general, all experiments are confined to one single language model, raising questions about cross-model transferability and robustness, limiting confidence in applicability on diverse model families and architectures.
- Insufficient Mechanistic Validation: While the paper proposes that column space preservation promotes forgetting resistance, it lacks direct quantitative evidence supporting this causal relationship. The theoretical claim requires empirical validation through metrics such as activation space similarity or representation overlap analysis before and after fine-tuning.


**Issues:**
- Incomplete Analysis of SVD Components: The method consistently employs all singular vectors without analyzing performance variations across different rank selections, missing opportunities for efficiency optimization and theoretical insights.
- Inadequate Table Documentation: Table captions are overly brief and fail to clearly explain notation and symbols (e.g., parameter L).
- Missing Routing Analysis: The paper lacks analysis of routing strategy sparsity patterns or expert collapse phenomena, which would be essential for genuine MoE evaluation. Though given the fundamental differences from traditional MoE experts, this may be less critical.

---

> ### Author Rebuttal · Authors · 2025-07-30
>
> Thanks for your comments. Below, we try to resolve your concerns one by one.
>
>
> **Q1: The novelty of the proposed method.**
>
> **A1:** We respectfully disagree with the comment that our work lacks novelty for the following reasons:
>
> **1. New methodology for constructing MoE:** To the best of our knowledge, our work makes the first attempt to convert an arbitrary model to an MoE architecture with the help of SVD decomposition. Different from existing MoE-based adaptation methods, which achieve MoE architectures by learning and routing external low-rank adapters, our MoORE construct orthogonal experts intrinsically from the SVD of a pretrained parameter matrix.
>
> **2. New methodology for achieving orthogonal experts efficiently:** Among existing MoE-based multi-task adaptation methods, OMoE is the only one considering the orthogonality of experts. However, it introduces orthogonality among experts’ outputs using the Gram-Schmidt orthogonalization algorithm, which requires high computational complexity. Our MoORE achieves orthogonal experts naturally by the SVD of the parameter matrix. The Housholder reflection imposed on each expert maintains their orthogonality while increasing their model capacity at the same time.
>
> **3. Considering the mitigation of task oblivion, which is seldom considered by existing methods:** Because of maintaining the column space of the original parameter matrix, MoORE has the ability to mitigate task oblivion. Existing multi-task adaptation methods mainly consider mitigating the conflicts among the downstream tasks. Few of them consider maintaining the original pre-training knowledge and mitigating the oblivion of pre-training tasks. Our work fills this blank.
>
> **4. Valuable routing analysis on MoE-based multi-task adaptation:** In the Appendix, we analyze MoORE’s routing weights across different tasks and provide valuable conclusions.
>
> **Q2: The differences between MoORE and existing SVD-based fine-tuning methods**
>
> **A2:** There are several key differences between MoORE and existing SVD-based fine-tuning methods like SVF [a]:
>
> **1. Methodology:** MoORE converts pre-trained models to MoE architectures in multi-task adaptation scenarios. On the contrary, without introducing any routing mechanism, existing SVD-based fine-tuning methods do not result in MoEs.
>
> **2. Application:** MoORE is designed to address multi-task learning problems of LLMs, mitigating task conflict and oblivion. SVF focuses on a few-shot image segmentation tasks. It does not consider learning a single model in simultaneous or sequential multi-task adaptation scenarios.
>
> [a] Singular value fine-tuning: Few-shot segmentation requires few-parameters fine-tuning. NeurIPS 2022.
>
>
> **Q3: The rationality and motivation for introducing the new concept "Model MoE-ization"**
>
> **A3:** Firstly, we must remind the reviewer that when MoE was first proposed in 1991, it was not sparsely activated. It was not until 2017 that sparsely activated MoE architectures were introduced, and the purpose of sparse activation was mainly to increase model capacity with high computational efficiency.
>
> We introduce the concept "Model MoE-ization" for the following reasons.
>
> 1) The community of multi-task adaptation, state-of-the-art methods introduce multiple LoRA adapters into pre-trained models and learn a router to adjust their significance based on input data or tasks, which actually converts the pre-trained model to an MoE architecture. In fact, the baselines like OMoE, LoRAMoE, MixLoRA, and MoSLD have explicitly claimed that they convert a pretrained model to an MoE architecture for multi-task adaptation. We introduce the concept "Model MoE-ization" to name this strategy, which matches well with its principle.
>
> 2) In fact, the development of MoE has always been about increasing model capacity to solve more complex tasks, while at the same time minimizing the loss in computational efficiency. Our SVD-based “Model MoE-ization” strategy is fully aligned with this philosophy. Our MoORE method alleviates task conflict and task forgetting, achieves better multi-task adaptation compared to other MoE-based baselines, and also reduces inference time overhead.
>
> Despite the MoE framing, the experimental evaluation completely omits comparisons with actual MoE methods, instead focusing on LoRA-based approaches. This fundamental disconnect between the conceptual narrative and empirical validation raises serious concerns about the paper's positioning and undermines the paper's central narrative.
>
> **Q4: The lack of MoE baselines**
>
> **A4:** **We respectfully disagree with this comment because all the baselines in our study are based on MoE architectures.** As shown in **A3**, the baselines like OMoE, LoRAMoE, MixLoRA, and MoSLD have explicitly claimed that they achieve multi-task adaptation by learning a model with an MoE architecture. The remaining methods (MTL-LoRA and HydraLoRA) leverage MoE architectures as well, based on their implementations.
>
> Note that it is reasonable to implement experts as low-rank adapters --- if each expert is full rank, the computational complexity will be too high and the model size will be too large. For the LLMs using MoE architectures, their experts are always simple and lightweight in practice.
>
>
> **Q5: The risk of limited generalizability due to the usage of task ID**
>
> **A5:** In our oblivion-resistance multi-task learning experiment (OR-MTL), we have considered scenarios where task IDs are unavailable - when new tasks come, we use the average of existing task ID embeddings as their ID embeddings. Specifically, after adapting LLaMA with MoORE on CSR-MTL, MoORE only learns the task IDs of the nine tasks in CSR-MTL. During evaluation on OR-MTL, we use the average of the nine task ID embeddings as the task ID embedding of each new task in OR-MTL. The results in Figure 1(c) and Figure 2 validate the effectiveness of this approach. We will add the content above in the revised version.
>
>
> **Q6: The lack of sensitivity analysis on $L$**
>
> **A6:** Table 3 has shown the robustness of our method to $L$: 1) When $L\in\{0, 2, 4, 8\}$, MoORE consistently outperforms all other baselines. 2) As $L$ varies, the results of MoORE do not change significantly—the variation is only 0.16%. These results demonstrate that the model is robust to $L$.
>
>
> **Q7: Add experiments on other model architectures.**
>
> A7: Following your suggestion, we take Qwen-2.5 7B as the backbone model and apply various MoE-based multi-task adaptation methods to it. Experimental results on CSR-MTL are shown below:
>
> |Qwen-2.5 7B|CSR-MTL|
> |-|-|
> |LoRA|84.35|
> |MixLoRA|85.53|
> |MoSLD|85.58|
> |HydraLoRA|85.87|
> |MTL-LoRA|85.73|
> |LoRAMoE |86.17|
> |MoORE L=0|86.30|
> |MoORE L=2|86.37|
> |MoORE L=4|86.38|
> |MoORE L=8|86.24|
> |MoORE L=16|**86.63**|
>
> We can find that 1) MoORE outperforms baselines consistently under different hyperparameter settings, and 2) the performance of MoORE is robust to the change of $L$. These results verify the robustness of our method and its compatibility with various model architectures
>
> **Q8: Insufficient mechanistic validation on the necessity of preserving the column space of the parameter matrix**
>
> **A8:** We have already stated in line 173 of the paper that “the work in [a] has shown that maintaining the column space of the weight matrix makes the adapted model inherit the ability of the pre-trained model better, mitigating the oblivion of the previously pre-trained task.” Our work further verifies this discovery and extends it in multi-task adaptation scenarios.
>
> [a] Bridging the gap between low-rank and orthogonal adaptation via householder reflection adaptation. NeurIPS, 2024.
>
>
> **Q9: The impacts of SVD components on efficiency**
>
> **A9:** Firstly, MoORE focuses on addressing task conflict and oblivion in multi-task adaptation. We admit that the model efficiency is important for multi-task adaptation, and we can further accelerate MoORE by various methods. However, as shown in Figure 1(d) and the analysis shown in Table 2, the current MoORE has achieved encouraging efficiency with strong performance. Further improvements in efficiency are out of the scope of this work.
>
> Secondly, as analyzed in Table 2, the number of singular vectors (equivalent to the number of experts) only partially contributes to computational complexity. We can improve efficiency by other methods, e.g., simplifying the router.
>
> Finally, since singular vectors contain pretrained knowledge, discarding any singular vectors will inevitably lead to a drop in model performance. We conducted experiments on CSR-MTL by retaining different proportions of singular values. The results are shown below.
>
> |MoORE|CSR-MTL|
> |-|-|
> |Top 25%, L=0|33.61|
> |Top 25%, L=8|35.32|
> |Top 50%, L=0|61.09|
> |Top 50%, L=8|65.28|
> |Top 75%, L=0|77.55|
> |Top 75%, L=8|78.78|
>
> Here, "Top 25%" means only the largest 25% of singular values are retained, reducing the number of experts to 25% of the original. The results confirm our claims --- even when 75% of the singular values and corresponding singular vectors are retained, MoORE’s performance drops significantly. Therefore, all singular values and singular vectors should be preserved.
>
> **Q10: Inadequate table documentation**
>
> **A10:** The notations and symbols appearing in each table have been defined/denoted in the related content, before the table is mentioned in the paper. For example, we have defined $L$ in Eq. (5). If possible, please indicate which table caption is too brief, and we will make improvements in the final version.
>
>
> **Q11: Missing routing analysis**
>
> **A11:** We have already provided an analysis of the distribution of routing weights across different tasks in the CSR-MTL dataset in Appendix C.3. Based on the mean and variance of the routing weights, we divided the nine tasks in CSR-MTL into three groups and presented our analysis and conclusions.
>
> **In summary, we hope the above responses can resolve your concerns and help re-evaluate our work.**

---

> ### Comment · Reviewer_eS7y · 2025-08-01
>
> Thank you for your response, but my major concern remains unresolved.
>
> Setting aside your excellent compelling narrative, I believe your work is not really different from various SVD tuning approaches.
> Almost all the advantages you emphasize stem from SVD tuning, and post-training with SVD decomposition is a common practice.
>
> Yet, you argue that this paper constitutes a new paradigm in MoE.
> If you had applied SVD to MoE and adapted it to common modern MoE models, that would have been an interesting innovation. But, you are fitting MoE-related concepts onto SVD tuning instead.
> You refer to the projection on singular values as "routing" and call the SVD orthogonal vectors "experts", but you still fail to explain why it is necessary to introduce the concept of MoE.
>
> In short, MoE is just your narrative; your approach is actually SVD tuning.
>
> (You've written really a good story, by the way.)

---

> ### Author Response · Authors · 2025-08-01
> **Round-2 Rebuttal: Further clarify the differences between MoORE and SVD fine-tuning**
>
> Thanks for your quick reply. However, for the following reasons, we respectfully disagree with your comment that MoORE is equivalent to SVD fine-tuning (SVF).
>
> **1) The difference in singular value adjustment in both training and inference:**
>
> - **SVF:** For a specific task, the singular values ($\boldsymbol{\sigma}$'s) are non-parametrically fine-tuned. During inference, regardless of the input data or task, the singular values remain fixed. Accordingly, SVF leads to a fine-tuned but fixed weight matrix, without changing model architectures.
> - **MoORE:** In multi-task scenarios, a router module is introduced to parameterize the components added to the singular values. After training, the singular values become $\boldsymbol{\sigma} + g(\boldsymbol{x}_n^{(k)})$, which adaptively change according to the input sample and task type. As a result, because the singular values are dependent on input, MoORE "MoE-izes" the original model, rather than returning a fine-tuned weight matrix.
>
> **2) The difference in model architecture (which might be the most important part):**
>
> - **SVF:** Since the singular values remain fixed regardless of the input, the SVD expression can be replaced by $\boldsymbol{W_{new}=US_{new}V^{\top}}$, which has the same shape as the original parameter matrix. Thus, the original model architecture is unchanged.
> - **MoORE:** Since the router adaptively adjusts the components added to the singular values based on the input and task type, the model architecture becomes an MoE-like architecture—the outer product of singular vectors multiplying learnable Householder reflections acts as experts, and the router determines the weights/activations of these experts. As shown in Eqs. (6, 7), this process cannot be implemented by a single, fixed matrix multiplication anymore, so the model architecture is fundamentally changed. This is why we emphasize that MoORE is a "Model MoE-ization" method, and why we compare MoORE with existing MoE-based multi-task adaptation methods. This is NOT just a conceptual or narrative difference, but a fundamental distinction at the level of model implementation.
>
> **3) The difference in trainable parameters:**
>
> - **SVF:** The trainable parameters are the singular values, whose number is fixed, determined by the number of singular values.
> - **MoORE:** The trainable parameters include the router and the $L$ Householder reflections, rather than the singular values. The number of trainable parameters can be flexibly adjusted by changing hyperparameter configuration (e.g., setting different $D_t$, $D_s$, and $L$ in different tasks).
>
> **4) The difference in application and problem:**
>
> - **SVF:** Existing SVD-based fine-tuning methods mainly focus on fine-tuning a model for a single task in a lightweight manner. These methods have not explored how to use SVD-based techniques to alleviate task conflict and oblivion in multi-task adaptation scenarios.
> - **MoORE:** To the best of our knowledge, our work makes the first attempt to use SVD to mitigate task conflict and oblivion in multi-task adaptation scenarios. Our work is also the first to systematically verify that maintaining expert orthogonality and column space of the weight matrix (achieved by our SVD-based method) provides feasible solutions to adapt a pre-trained model with enhanced resistance to task conflict and oblivion.
>
> **5) The role of our work in connecting SVD with MoE, and the necessity of emphasizing "Model MoE-ization":**
>
> - We are NOT simply creating the concept of "Model MoE-ization" and fitting it onto SVD fine-tuning. As the above analysis shows, existing SVF methods cannot convert any model layer into an MoE architecture (which is why these methods do not connect SVD with MoE). In contrast, our work does not apply SVD to an existing MoE-based model, but instead, uses SVD to convert an arbitrary linear layer into a MoE layer (i.e., model MoE-ization) for multi-task adaptation.
>
>
> **6) The summary of the above differences.**
>
> - Even if, for the sake of argument, our MoORE is considered a form of SVD fine-tuning, it is an **(1) adaptive** SVD fine-tuning method **(2) oriented to multi-task adaptation** and **(3) with resistance to task conflict and oblivion**, which leads to **(4) the change of model architecture** after adaptation. The abundance of these qualifiers has clearly demonstrated the significant differences between MoORE and SVF.
>
> **7) The logic behind the comment is questionable when it is applied to evaluate other methods.**
>
> - Finally, if we follow your logic, shall we also consider that the baselines used in our paper, such as HydraLoRA, LoRAMoE, and MixLoRA, are merely narrative variations of LoRA? Are they essentially just LoRA? Are these works just "good stories"? In our opinion, such an arbitrary viewpoint is unreasonable.
>
> In conclusion, we sincerely hope you will reconsider your evaluation of our work.

---

> > ### Comment · Reviewer_eS7y · 2025-08-03
> >
> > Dear Authors,
> >
> > After my careful review and re-review of your paper, I maintain that your proposed method does not sufficiently differentiate from SVD tuning or singular value tuning (as highlighted in my initial review).
> >
> > While your response provides context, it does not demonstrate innovative changes in your proposed framework to distinguish it from existing methods.
> >
> > Given the standards of NIPS for contributions, I find that the current work does not meet the threshold for acceptance.
> >
> > I will keep my score but appreciate your efforts.

---

> > > ### Author Response · Authors · 2025-08-09
> > >
> > > Dear Reviewer eS7y,
> > >
> > > We have provided a general response (in https://openreview.net/forum?id=g42mGfR6We&noteId=KRhhBSk7zh), in an attempt to resolve your remaining concerns for a final time. Besides the SVF you mentioned, we further compare our MoORE with four more existing SVD-based fine-tuning methods systematically.
> > >
> > > We hope our response can avoid any possible misunderstandings and help re-evaluate our work. Thanks again for your consideration.

---

### Official Review · Reviewer_P3D7 · 2025-07-03

**Clarity:** 3
**Significance:** 3
**Originality:** 3
**Rating:** 5
**Confidence:** 4

**Summary:**

Mixture of Experts is now a popular paradigm for multi-task adaptation of transformers. In this approach an adapter is assigned/learned for each task and a learnable router selects the appropriate adapter for the given task. This achieves parameter efficient fine-tuning. An important drawback of this approach is destructive interference and catastrophic forgetting. A number of solutions has been suggested for to solve this problem. Many of them lack elegance and a theoretical foundation.

The present paper proposes an simple yet effective approach to solve this problem. It constructs a stack of "weak"experts using orthogonal factorization of the foundation weight matrix. A router is learned for choosing among these simple experts. The method is found to be effective on benchmark tasks.

**Questions:**

1. Is it possible to provide a set of formulations in terms of the underlying linear algebra operations of the proposed method vis-a-vis other MoE-zation techniques for PEFT?

2. How does the proposed method compare with other state-of-art PEFT techniques in terms of efficiency on multi-task benchmarks?

**Ethical Concerns:**

["NO or VERY MINOR ethics concerns only"]

**Limitations:**

The proposed approach has a great deal of similarity with the eigenstructure assignment problem in linear algebra. It would be beneficial to study this relation in order to have a better theoretical understanding of the interference problem.Often "regularization functionals"are used in eigenstructure assignment. Such functionals may be useful for PEFT as well.

**Paper Formatting Concerns:**

No formatting concerns.

**Quality:**

3

**Strengths And Weaknesses:**

The paper presents an excellent application of linear algebra tools for mixture of experts formulation of the parameter efficient fine-tuning process. The results demonstrate the effectiveness of this relatively simple approach.

A number of very similar approaches to MoE-ization is existent in literature. Many of them are compared in the experimental results section. They are also qualitatively described. A more mathematical formulation of the "weak" MoE approach as opposed to exisitng ones using routers in the light of linear algebra concepts might be possible and will definitely add more value to the topic/ It will definitely affirm the novelty of the proposed technique.

---

> ### Author Rebuttal · Authors · 2025-07-30
>
> Thank you very much for your appreciation of our work. For your concerns, we provide our answer below.
>
>
> **Q1: The formulations of other MoE-zation methods for PEFT and the comparisons for various methods mathematically.**
>
> **A1:** In our submission, **the formulation of most existing MoE-zation methods has been shown in Eq. (2), and the formulation of our MoORE is given in Eq. (6).** We paste the equations below for a quick review.
>
> Given a pre-trained parameter matrix $\mathbf{W}$, whose SVD is $\mathbf{U}\text{diag}(\boldsymbol{\sigma})\mathbf{V}^{\top}$ and an input of the $k$-th task $\mathbf{x}^{(k)}$, the formulations of MoORE and baselines are shown below:
>
> MoORE:
>
> $\mathbf{y} =\mathbf{U}(g(\mathbf{x}^{k})+\boldsymbol{\sigma})\mathbf{V}^{\top}\mathbf{WHx}^{(k)}=\mathbf{WHx}^{(k)}+\sum_{d=1}^{D}g_d(\mathbf{x}^{(k)})(\mathbf{u}_d\mathbf{v}_d^{\top}\mathbf{H})\mathbf{x}^{(k)}$,
>
> where $\mathbf{H}$ is a learnable Householder reflection matrix, and $g$ is the router determining the weight of each expert $\mathbf{u}_d\mathbf{v}_d^{\top}\mathbf{H}$.
>
> Baselines:
>
> $\mathbf{y} =\mathbf{Wx}^{(k)}+\sum_{m}g_m(\mathbf{x}^{(k)})\mathbf{B}_m\mathbf{A}_m\mathbf{x}^{(k)}$,
>
> where $\mathbf{B}_m$ and $\mathbf{A}_m$ are low-rank matrices of the $m$-th expert, and $g_m$ is the weight of the expert determined by a router.
>
> In Table 1, we have shown the implementation details of the methods, corresponding to the specific configurations of the routers and experts shown in Eqs. (2) and (6).
>
> In Lines 161-176, we have explained the differences between MoORE and the existing methods from the perspective of linear algebra. Following your suggestion, we emphasize the differences below for a quick review.
>
> **1) The maintenance of orthogonality mitigates task conflict:** MoORE leverages $D$ rank-1 and orthogonal experts extracted from the given parameter matrix. The orthogonality eliminates the information redundancy across our experts. In contrast, the baselines learns $M$ LoRA adapters as experts ($M\ll D$), without any orthogonality constraints. As a result, for each baseline method, the output spaces of its experts overlap, and accordingly, different experts may output similar latent representations with redundant information. Eliminating the information redundancy helps mitigate task conflict, i.e., the orthogonal experts of our model can serve for different tasks.
>
> **2) The maintenance of output space mitigates task oblivion:** Because MoORE leverages experts extracted from the given parameter matrix, it does not change the output space of the model, as shown in Eq. (8). In PEFT-based single-task adaptation, the work like HRA [a] has shown that maintaining the range of parameter matrix (the output space of model) is important for avoiding losing the model capability obtained in the pre-training phase. Our work further extends this claim to both simultaneous and sequential multi-task adaptation scenarios.
>
> [a] Bridging the gap between low-rank and orthogonal adaptation via householder reflection adaptation. NeurIPS 2024.
>
> **Q2: The efficiency of MoORE**
>
> **A2:** In our submission, **Figure 1d shows a comparison of various methods on inference efficiency.** Among them, MoORE is only slightly less efficient than MixLoRA, but outperforms all other baselines. This demonstrates the efficiency of MoORE in practice. Besides, we have analyzed the computational complexity of various methods in Table 2 and Lines 177-191, which demonstrates in theory that the complexity of MoORE is comparable to the baselines.
>
> **Q3: The connection to the eigenstructure assignment problem**
>
> **A3:** Thanks for your insightful suggestion! We think connecting MoORE to the eigenstructure assignment problem is interesting. Although it is out of the scope of this work, we will definitely explore their connections in the near future. In the final version, we will discuss the potential connection of our work with eigenstructure assignment in the conclusion section and outline it as a future direction.
>
> Note that, in Appendix C.3, we have provided a detailed empirical analysis on the singular spectrum of the parameter matrix (i.e., the weight of experts determined by the route) for different tasks. Some empirical patterns have been found and discussed. In our opinion, this preliminary analysis makes the first step towards exploring the theory behind MoORE.
>
> In summary, we hope the above responses can resolve your concerns thoroughly and further enhance your confidence to support our work in the following decision phase.

---

> > ### Comment · Reviewer_P3D7 · 2025-08-02
> >
> > Thank you. The clarifications on the methodology is well appreciated. Some of my doubts on the reasons for effectiveness of the proposed methodology is clear now. I appreciate the elegance of your approach.

---

> > > ### Author Response · Authors · 2025-08-03
> > >
> > > Thank you so much for your appreciation of our work. It would be nice  if you could continue to support our work during the upcoming discussion phase.

---

> > ### Comment · Reviewer_P3D7 · 2025-08-09
> >
> > We rate the work quite high. The rebuttals have made several points clear. Multi-task adaptation continues to be an intriguing topic. We urge the authors to reiterate the importance of the algorithm in the context of several work in this direction.

---

> > > ### Author Response · Authors · 2025-08-09
> > >
> > > Thanks for your constructive suggestions. We totally agree that multi-task adaptation is critical for the application of large-scale foundation models, which motivates us to develop MoORE. As we mentioned in the above general response (https://openreview.net/forum?id=g42mGfR6We&noteId=KRhhBSk7zh), besides the LoRA-based baselines, we will discuss more relevant works (e.g., existing SVD-based fine-tuning methods) and further clarify our contributions in the revised paper. Again, thank you very much for your appreciation of our work.

---

### Author Response · Authors · 2025-08-05
**A General Response before the End of the Author-Reviewer Discussion Phase**

Dear Reviewers, AC, SAC, and PC,

We thank all reviewers for their comments and replies. Currently, the reviewer eS7y is the only one evaluating our work negatively because of the concerns about **1) the differences between our MoORE and existing SVD-based fine-tuning methods**, and **2) the necessity of introducing the concept "Model MoE-ization."** Although we have tried our best to resolve these two concerns in our response (see https://openreview.net/forum?id=g42mGfR6We&noteId=7Ap4w4gEKO), we failed to change his/her mind, unfortunately.

**To avoid any potential misunderstandings, we provide this general response for your consideration in the following discussion and decision phases.**

**1) The differences between MoORE and SVD-based fine-tuning**

Besides the SVF suggested by Reviewer eS7y, we collected four more SVD-based methods and compared them with MoORE.

1. **SVF:** Singular value fine-tuning: Few-shot segmentation requires few-parameters fine-tuning. NeurIPS 2022.
2. **SVDiff:** Svdiff: Compact parameter space for diffusion fine-tuning. ICCV 2023.
3. **SVFT:** Svft: Parameter-efficient fine-tuning with singular vectors. NeurIPS 2024.
4. **KaSA:** KaSA: Knowledge-Aware Singular-Value Adaptation of Large Language Models. ICLR 2025.
5. **SVFCL:** Singular Value Fine-tuning for Few-Shot Class-Incremental Learning. arXiv 2025.

Denote a parameter matrix as $\mathbf{W}$, its SVD as $\mathbf{U}\text{diag}(\mathbf{\sigma})\mathbf{V}^{\top}$, and its low-rank estimation as $\mathbf{W}_{lr}$. The table below summarizes the differences between MoORE and these methods.

|Method|Model in Training|Learnable Parameters|Model in Inference|Multi-Task Learning|Consider Task Conflict|Consider Task Oblivion|
|-|-|-|-|-|-|-|
|MoORE| $\mathbf{WH}+\sum_{d=1}^{D}g_d(\mathbf{x}^{(k)})(\mathbf{u}_d\mathbf{v}_d^{\top}\mathbf{H})$|Parametric $g(\cdot)$ and  Householder reflections $\mathbf{H}$|An MoE: $\mathbf{WH}+\sum_{d=1}^{D}g_d(\mathbf{x}^{(k)})(\mathbf{u}_d\mathbf{v}_d^{\top}\mathbf{H})$, where $\mathbf{WH}$ and $\mathbf{Hv}_d$ are computed in advance |Yes|Yes, by orthogonality of experts|Yes, by maintaining column space of $\mathbf{W}$|
|SVF|$\mathbf{U}\text{diag}(\mathbf{\sigma})\mathbf{V}^\top$|Nonparametric $\mathbf{\sigma}$|Original architecture: $\mathbf{W}_{new}$|No|No|No|
|SVDiff|$\mathbf{U}\text{diag}(\text{ReLU}(\mathbf{\sigma+\delta})) \mathbf{V}^\top$|Nonparametric $\mathbf{\delta}$|Original architecture: $\mathbf{W}_{new}$|No|No|No|
|SVFT|$\mathbf{U}(\text{diag}(\mathbf{\sigma}) + \mathbf{M})\mathbf{V}^\top$|Nonparametric $\mathbf{M}$|Original architecture: $\mathbf{W}_{new}$|No|No|No|
|KaSA|$\mathbf{W}_{lr}+\Delta\mathbf{U}\text{diag}(\Delta\mathbf{\sigma})\Delta\mathbf{V}^\top$|Nonparametric $\Delta\mathbf{U}$, $\Delta\mathbf{\sigma}$, $\Delta\mathbf{V}^\top$|Original architecture: $\mathbf{W}_{new}$|No|No|No|
|SVFCL|$\mathbf{U}\text{diag}(\mathbf{\sigma}+\sum_{i=0}^{t}\mathbf{\sigma}_i)\mathbf{V}^\top$|Nonparametric $\mathbf{\sigma}_i$'s|Original architecture: $\mathbf{W}_{new}$|Yes, in the incremental learning scenario|No|Partially discussed|

- **In the aspect of architecture, MoORE is the only method that learns a parametric router $g$ to adjust singular values adaptively according to input, leading to an MoE architecture for inference**. In MoORE, each expert is constructed by the outer product of singular vectors and the learnable Householder reflections.

- **In the aspect of application, only MoORE and SVFCL consider multi-task adaptation. MoORE is applicable for both simultaneous and sequential multi-task adaptation, while SVFCL only considers sequential/incremental multi-task learning.** Moreover, MoORE makes the first attempt to connect task conflict and oblivion to the orthogonality of experts and the column space of the parameter matrix.

**2) The necessity of the concept Model MoE-ization**

As shown in the table, **our MoORE is the only method leading to an MoE architecture for inference, which converts a non-MoE model to an MoE.** Compared to maintaining the original model architecture, the model MoE-ization can increase the flexibility and scalability of the model when dealing with multiple tasks --- the singular values (i.e., the activation of experts) are adjusted according to the input and tasks, which naturally help mitigate task conflict and oblivion. In other words, "Model MoE-ization" precisely describes the technical route of our method --- it is not narrative but provides a promising solution (overlooked by the community to some degree) to conflict- and oblivion-resistant multi-task adaptation.


In summary, our work is significantly different from existing SVD-based methods in both methodology and application, and the concept "Model MoE-ization" is meaningful in practice. **We will add the above analysis and references in the revised paper, but the concerns on these two points should not be the evidence for rejection.** Thanks in advance for your consideration.

---

### Note · Authors · 2025-08-11

Dear Reviewers, AC, SAC, and PC,

We thank all reviewers for their comments and replies.
For only two negative comments proposed by Reviewer eS7y, we have provided our answers in the following two responses in the past few days.

1. **The general response** in https://openreview.net/forum?id=g42mGfR6We&noteId=KRhhBSk7zh

2. **The specific response to Reviewer eS7y** in https://openreview.net/forum?id=g42mGfR6We&noteId=7Ap4w4gEKO.

In this final remark, we further summarize our response in short.

**Q1. The differences between MoORE and existing SVD-based fine-tuning methods.**

**A1:** In our responses, we have compared our MoORE with five recent and representative SVD-based fine-tuning methods systematically, clearly showing that MoORE is different from these competitors in 1) the parametrization strategy, 2) the model architecture in the inference phase, 3) the application scenario, and 4) the solutions to the key challenges associated with the application scenario.
In our opinion, the comparisons have demonstrated the contribution and novelty of our work.

**Q2. The necessity of introducing the concept "Model MoE-ization"**

**A2:** As shown in our responses, our work provides a new SVD-based method to convert pre-trained models to MoE architectures in multi-task adaptation scenarios. Because of maintaining the orthogonality of experts and the output space, our work leads to a new solution to achieve conflict- and oblivion-resistant multi-task adaptation, which has a chance to overcome not only parameter-efficient fine-tuning but also full-parameter fine-tuning. We introduce the concept "model MoE-ization" to 1) highlight the principle of our method and 2) indicate the significance of the methodology for the community.

Again, we hope our responses can resolve the concerns and avoid any misunderstandings in the following decision phase.
Thank you very much for your consideration.

Regards,

Authors of Submission 12046

---

### Decision · Program_Chairs · 2025-09-17

**Decision:**

Accept (poster)

**Comment:**

This paper proposes MoORE, a novel approach to convert pre-trained dense models into orthogonal rank-one “experts” combined through a learnable router for multi-task adaptation. Through SVD on the weight matrix of a pre-trained model, the proposed approach maintains the experts’ orthogonality and the column space of the original weight matrix which provides a simple yet effective solution to address task conflict and oblivion challenges in multi-task adaptation. At the same time, the method is computationally efficient with minimum learnable parameters. The authors provide a clear theoretical motivation and validate the approach using diverse commonsense reasoning and natural language understanding benchmarks.

While one reviewer raised concerns about novelty and framing, the rebuttal offers detailed comparisons to representative SVD-based competitors, and discussion of related MoE-style methods, clarifying MoORE’s unique positioning. Experimental results validate the strong performance compared to baselines, supporting the method’s practical value. Upon carefully reading the reviews and authors' responses to these concerns, it appears the concerns are more of different views of the approach, and I believe the comparison with competitive baselines are clear in terms of both methodology and empirical results. I'd suggest adding these discussion to the final paper.

Overall, given its simplicity, theoretical grounding, and empirical effectiveness, I believe MoORE will be a valuable contribution to the efficient adaptation of large language models. I'd suggest accepting the paper.